# Combinatorial regulation of the balance between dynein microtubule end accumulation and initiation of directed motility

Rupam Jha[†], Johanna Roostalu, Nicholas I Cade, Martina Trokter[‡] & Thomas Surrey[*] [iD]

## Abstract

Cytoplasmic dynein is involved in a multitude of essential cellular functions. Dynein's activity is controlled by the combinatorial action of several regulatory proteins. The molecular mechanism of this regulation is still poorly understood. Using purified proteins, we reconstitute the regulation of the human dynein complex by three prominent regulators on dynamic microtubules in the presence of end binding proteins (EBs). We find that dynein can be in biochemically and functionally distinct pools: either tracking dynamic microtubule plus-ends in an EB-dependent manner or moving processively towards minus ends in an adaptor protein-dependent manner. Whereas both dynein pools share the dynactin complex, they have opposite preferences for binding other regulators, either the adaptor protein Bicaudal-D2 (BicD2) or the multi-functional regulator Lissencephaly-1 (Lis1). BicD2 and Lis1 together control the overall efficiency of motility initiation. Remarkably, dynactin can bias motility initiation locally from microtubule plus ends by autonomous plus-end recognition. This bias is further enhanced by EBs and Lis1. Our study provides insight into the mechanism of dynein regulation by dissecting the distinct functional contributions of the individual members of a dynein regulatory network.

**Keywords** dynactin; dynamic microtubules; human dynein; microtubule plus-end tracking; regulation of dynein motility

**Subject Categories** Cell Adhesion, Polarity & Cytoskeleton; Membrane & Intracellular Transport

The EMBO Journal (2017) 36: 3387–3404

## Introduction

Cytoplasmic dynein-1 (here called dynein) is the major minus-end-directed microtubule motor in most eukaryotes (Vale, 2003). It is involved in a variety of cellular functions ranging from organelle transport (Allan, 2011) to spindle pole focussing (Merdes *et al*, 2000), removal of checkpoint proteins from kinetochores (Griffis *et al*, 2007) and spindle positioning (McGrail & Hays, 1997). Dynein is a 1.4 MDa protein complex consisting of two copies of six different subunits that assemble into a tail domain from which two motor domains protrude (Vallee *et al*, 1988; Vale, 2003; Pfister *et al*, 2006). The motile properties of metazoan dynein depend strongly on its interaction with a variety of regulatory proteins whose mechanisms of action and their combinatorial interplay are poorly understood (Vallee *et al*, 2012; Cianfrocco *et al*, 2015).

The major regulator of dynein that is required for most of its cellular functions is dynactin, a ~ 1.2 MDa protein complex that consists of various copies of eleven different protein subunits (Karki & Holzbaur, 1999; Schroer, 2004; Urnavicius *et al*, 2015). Despite being the key dynein regulator, dynactin itself interacts only weakly with dynein (McKenney *et al*, 2014; Schlager *et al*, 2014). Additional adaptor proteins that recruit dynein to cargoes stabilise this interaction, leading to ternary complex formation and activation of processive dynein motility (McKenney *et al*, 2014; Schlager *et al*, 2014). Ternary complex formation is thought to release dynein from its autoinhibited state, possibly by separating the two motor domains (Chowdhury *et al*, 2015; Urnavicius *et al*, 2015; Carter *et al*, 2016).

There are several such adapter proteins that link dynein/dynactin to different cargoes like organelles and kinetochores or to the cell cortex (Kardon & Vale, 2009; McKenney *et al*, 2014). One example is the metazoan-specific adapter protein Bicaudal-D2 (BicD2) that is critical for bidirectional transport of mRNA particles and contributes to the positioning of the endoplasmic reticulum, the Golgi apparatus and the nucleus (Bullock & Ish-Horowicz, 2001; Hoogenraad *et al*, 2001; Hoogenraad & Akhmanova, 2016). The N-terminal coiled-coil of BicD2 mediates the interaction between the dynein tail and dynactin, which is crucial for processive minus-end-directed dynein motion (McKenney *et al*, 2014; Schlager *et al*, 2014; Chowdhury *et al*, 2015; Urnavicius *et al*, 2015), whereas the C-terminal part of BicD2 interacts with Rab receptors on cargo membranes (Hoogenraad *et al*, 2001; Hoogenraad & Akhmanova, 2016).

Despite being a minus-end-directed motor, dynein is also well known to accumulate at the plus ends of growing microtubules in

The Francis Crick Institute, London, UK

*Corresponding author. Tel: +44 2037962044; E-mail: thomas.surrey@crick.ac.uk

[†]Present address: Discovery Sciences, Innovative Medicines and Early Development, AstraZeneca, Cambridge, UK

[‡]Present address: Centre for Therapeutics Discovery, MRC Technology, SBC Open Innovation Campus, Stevenage, UK

cells (Kardon & Vale, 2009). This interaction is thought to enable dynein to initiate cargo transport at microtubule plus ends (Vaughan *et al*, 2002; Egan *et al*, 2012; Moughamian & Holzbaur, 2012) and to facilitate dynein-mediated interactions between microtubule plus ends and other cellular structures like the cell cortex (McGrail & Hays, 1997) or the kinetochores (Faulkner *et al*, 2000). Different pathways are known to be responsible for microtubule plus-end accumulation of dynein. Kinesin-dependent transport has been observed in fungi and neurons of metazoans, whereas EB1 family protein (EB)-dependent end tracking has been described for non-neuronal cultured mammalian cells (Carvalho *et al*, 2004; Moughamian *et al*, 2013; Roberts *et al*, 2014).

End binding proteins autonomously bind growing microtubule plus-end regions by recognising the nucleotide state of freshly added tubulins (Bieling *et al*, 2007; Maurer *et al*, 2011). EBs recruit a multitude of other plus-end tracking proteins (Akhmanova & Steinmetz, 2015), including dynactin, which is additionally required for plus-end tracking of dynein in cells (Vaughan *et al*, 1999; Ligon *et al*, 2003; Dixit *et al*, 2008). The critical dynactin subunit for plus-end tracking is p150[Glued] (called p150 here). Its N-terminal CAP-Gly domain protrudes from the shoulder of the dynactin complex (Chowdhury *et al*, 2015; Urnavicius *et al*, 2015) and binds directly to microtubules (Culver-Hanlon *et al*, 2006; Peris *et al*, 2006; Ross *et al*, 2006), as well as to EBs (Honnappa *et al*, 2006). *In vitro* reconstitution experiments with a p150 fragment showed that the p150 CAP-Gly domain and the first coiled-coil of p150, which interacts with the intermediate chain of the dynein complex (Karki & Holzbaur, 1995; Vaughan & Vallee, 1995; King *et al*, 2003), were sufficient for mediating EB-dependent end tracking of the human dynein complex (Duellberg *et al*, 2014). The regulation of the balance between dynein microtubule end tracking and processive motility was, however, not studied in these experiments as this requires the presence of the entire dynactin complex. Furthermore, in a recent cryo-electron microscopy structure of the dynactin complex, both the p150 CAP-Gly domain and the first coiled-coil appear buried in a groove of the dynactin shoulder, raising the question of whether p150 in the context of the entire dynactin complex can mediate EB-dependent dynein end tracking at all, or if additional factors might be needed to release a potential autoinhibition of dynactin (Urnavicius *et al*, 2015).

Dynactin-dependent plus-end localisation of dynein has been demonstrated to be involved in the control of transport initiation in distal neurites (Moughamian *et al*, 2013; Nirschl *et al*, 2016). In non-polarised cells, the role of dynactin in transport initiation is, however, not so well understood (Kim *et al*, 2007; Dixit *et al*, 2008). From a mechanistic point of view, it is unclear whether EB-recruited dynactin can contribute to initiating transport, thereby promoting transport initiation preferentially from microtubule plus ends. Alternatively, it is conceivable that dynein tracking microtubule ends and dynein initiating processive runs belong to separate dynein pools.

Another major question concerns the mechanism by which the dynein regulator Lissencephaly-1 (Lis1) controls the balance between plus-end tracking of dynein versus initiation of minus-end-directed motility. Lis1 has been reported to regulate both initiation of minus-end-directed motility (Egan *et al*, 2012; Splinter *et al*, 2012; Moughamian *et al*, 2013) and plus-end tracking of dynein (Coquelle *et al*, 2002; Splinter *et al*, 2012) in cells. However, Lis1

was not required for dynein end tracking in a minimal *in vitro* reconstitution (Duellberg *et al*, 2014), raising the question as to why Lis1 is needed for dynein end tracking in cells. Lis1 is a homo-dimeric 45 kDa protein that binds directly to the dynein motor domain (Mateja *et al*, 2006; Kardon & Vale, 2009), and was reported to induce a more strongly microtubule-bound state of dynein (Yamada *et al*, 2008; McKenney *et al*, 2010; Torisawa *et al*, 2011), thereby increasing the force produced by dynein (McKenney *et al*, 2010; Reddy *et al*, 2016), and slowing down microtubule transport by surface-immobilised dynein motors (Yamada *et al*, 2008; Torisawa *et al*, 2011; Wang *et al*, 2013).

To improve our understanding of how the combined action of various dynein regulators controls dynein behaviour, *in vitro* reconstitutions are needed that allow the simultaneous study of processive motility and microtubule plus-end tracking of dynein to dissect the respective roles of the various regulators.

Here, we perform *in vitro* reconstitution experiments probing the mechanism of the combinatorial regulation of recombinant human dynein by three human dynein regulators—dynactin, BicD2 and Lis1—in the presence of EB proteins. We find that the dynactin complex alone can mediate EB-dependent end tracking of dynein, thereby biasing dynein localisation towards microtubule ends. Despite competition for dynein binding, Lis1 and BicD2 cooperate to initiate processive motility. Plus-end tracking dynein and processively moving dynein seem to comprise two distinct dynein pools defined by distinct interaction partners. Finally, we find that dynactin has the previously unknown capacity to initiate processive dynein motility preferentially from microtubule plus ends, independent of EB proteins. These reconstitutions provide insight into the basic molecular mechanism of biased transport initiation of dynein from microtubule plus ends.

## Results

To investigate the behaviour of human cytoplasmic dynein and dynactin on dynamic microtubules, we purified both protein complexes. GFP-tagged human dynein was expressed and purified from insect cells as described (Trokter *et al*, 2012; Schlager *et al*, 2014) (Materials and Methods). The human dynactin complex was purified from HeLa S3 cell cultures using a new method based on BicD2 affinity chromatography followed by ion exchange chromatography (Materials and Methods). Purity of both complexes was demonstrated by SDS–PAGE, and the subunit composition of the dynactin complex was verified by mass spectrometry (Fig EV1A). To be able to study simultaneously dynamic microtubule end tracking and processive motility initiation of dynein/dynactin, we adjusted the conditions of previous *in vitro* reconstitution experiments, using here a buffer that was intermediate in ionic strength compared to previous dynein motility studies on static microtubules (Trokter *et al*, 2012; McKenney *et al*, 2014; Schlager *et al*, 2014) and microtubule end tracking experiments (Duellberg *et al*, 2014; Roberts *et al*, 2014) (Materials and Methods).

We first investigated whether the purified human dynactin complex can recruit dynein to dynamic microtubule plus ends in the presence of purified EB1 (Fig EV1B). These experiments were intended to test whether in the absence of other dynein/dynactin regulators, the N-terminal part of the dynactin component p150 is

undocked from the full dynactin complex and available for EB binding, and whether the reported weak interaction between dynein and dynactin in the absence of cargo adaptors (Chowdhury *et al*, 2015; Urnavicius *et al*, 2015) can support efficient dynein microtubule end tracking. Using time-lapse dual colour total internal reflection fluorescence (TIRF) microscopy, we observed dynamic microtubules growing from immobilised GMPCPP-stabilised microtubule "seeds" in the presence of 10 nM GFP-dynein, 20 nM dynactin, 20 nM purified EB1 and 17.5 μM Alexa568-tubulin. GFP-dynein accumulated strongly on growing microtubule plus and minus ends (Fig 1A and B and Movie EV1), typical for recruitment depending on EB1 family proteins (EBs). Additionally, some diffusive and static GFP-dynein binding events were observed all along the microtubules. GFP-dynein was also enriched on shrinking microtubule ends after microtubules had switched to depolymerisation (Fig 1A and B).

Using an Alexa647-labelled version of the SNAP-tagged EB1 homologue EB3 (Fig EV1B), we observed that GFP-dynein indeed colocalised with the EB3 at growing microtubule ends in the presence of dynactin (Fig 1C) in an ATP-independent manner (Fig 1G). Omitting either dynactin (Fig 1D) or EB protein (Fig 1E) from the assay strongly reduced end tracking of GFP-dynein as evidenced by the averaged GFP-dynein intensity profiles at the plus ends (Fig 1F). These results demonstrate that the human dynactin complex is sufficient to recruit the dynein complex to EBs at growing microtubule ends. Replacing human dynactin by purified neuronal pig brain dynactin (Fig EV1B) resulted in similar microtubule end tracking behaviour of dynein (Fig EV2), however with reduced efficiency, probably due to a fraction of neuronal dynactin containing the p150 isoform p135 (Fig EV1B, left) that lacks the N-terminal CAP-Gly domain (Tokito *et al*, 1996). These results indicate that both purified dynactin complexes can form a link between EBs and dynein, when the p150 subunit is present. Together, these results suggest that in the context of the entire dynactin complex, the CAP-Gly domain of p150 can be exposed, to allow end tracking of dynein without the requirement of other regulators. These findings go beyond previous work with a p150 fragment (Duellberg *et al*, 2014) providing insight into the ability of the dynactin complex to mediate dynein end tracking.

The interaction between dynein and dynactin is stabilised by a variety of cargo adaptors, such as Bicaudal-D2 (BicD2), which activates processive dynein motion (McKenney *et al*, 2014; Schlager *et al*, 2014; Hoogenraad & Akhmanova, 2016). It is not known how this stabilisation and activation of processive motion influences the

end tracking behaviour of dynein. To address this question, we purified an N-terminal fragment of the human cargo adaptor protein BicD2 (BicD2-N) (Fig EV1B). This fragment lacks the ability of autoinhibition (Hoogenraad *et al*, 2001) and has been shown to trigger processive dynein/dynactin motility (Splinter *et al*, 2012; McKenney *et al*, 2014; Schlager *et al*, 2014). Addition of 200 nM Alexa647-labelled SNAP-tagged BicD2-N (Alexa647-BicD2-N) to a reconstitution assay with dynamic microtubules, dynein, dynactin and EB1 leads now to the appearance of processive and lattice-bound GFP-dyneins in addition to plus-end accumulated dynein (Fig 2A). This is one of the first observations of processive human dynein motility on dynamic microtubules using purified proteins (Baumbach *et al*, 2017).

BicD2-N colocalised mainly with processive GFP-dynein (Fig 2A), as expected for the ternary complexes consisting of dynein, dynactin and BicD2-N (DDB) (McKenney *et al*, 2014; Schlager *et al*, 2014). Interestingly, BicD2-N did not colocalise with plus-end tracking GFP-dynein (Fig 2A and B). Velocity and run length of processive DDB particles and the relative fractions of processive, diffusive and static DDB binding events on dynamic microtubules were similar to those reported for static microtubules (Fig EV3) (McKenney *et al*, 2014; Schlager *et al*, 2014). Hence, the DDB complex was functional in our assay condition that supports microtubule dynamics.

Adding a large excess of BicD2-N (5 μM) to the end tracking experiment led to a strong reduction in EB1-dependent end tracking of GFP-dynein (Fig 2C and D, Movie EV2), as evidenced by averaged GFP-dynein intensity profiles at dynamic microtubule plus ends (Fig 2E). At the same time, the increased concentration of BicD2-N induced more processive runs (Fig 2D). These results show that DDB complex formation and processive motility are apparently incompatible with EB-mediated microtubule plus-end tracking of dynein.

This led us to ask whether this incompatibility is due to DDB particles processively leaving the EB-decorated end, which would predict that EBs would promote the initiation of runs from microtubule plus ends. To test this, we measured the relative initiation frequency of processive DDB runs in the presence and absence of EB1 as a function of the distance from the growing microtubule plus end (Fig 3). Surprisingly, we observed a strongly increased initiation probability in the growing plus-end region even in the absence of EB1 (Figs 3A and B, black bars and Fig EV4A, left): the initiation probability was 0.33 within the first 360 nm from the microtubule end compared to an average of 0.05 per 360 nm segment anywhere

Figure 1.

on the microtubule. This strong bias of the initiation probability was further increased to some extent by the additional presence of EB1 (from 0.33 to 0.42 within the first 360 nm, Figs 3B, red bars and Fig

EV4A, right) probably by increasing the local concentration of dynein/dynactin at plus ends (Fig 3C). The strong bias of initiations of processive runs from the microtubule plus-end region was

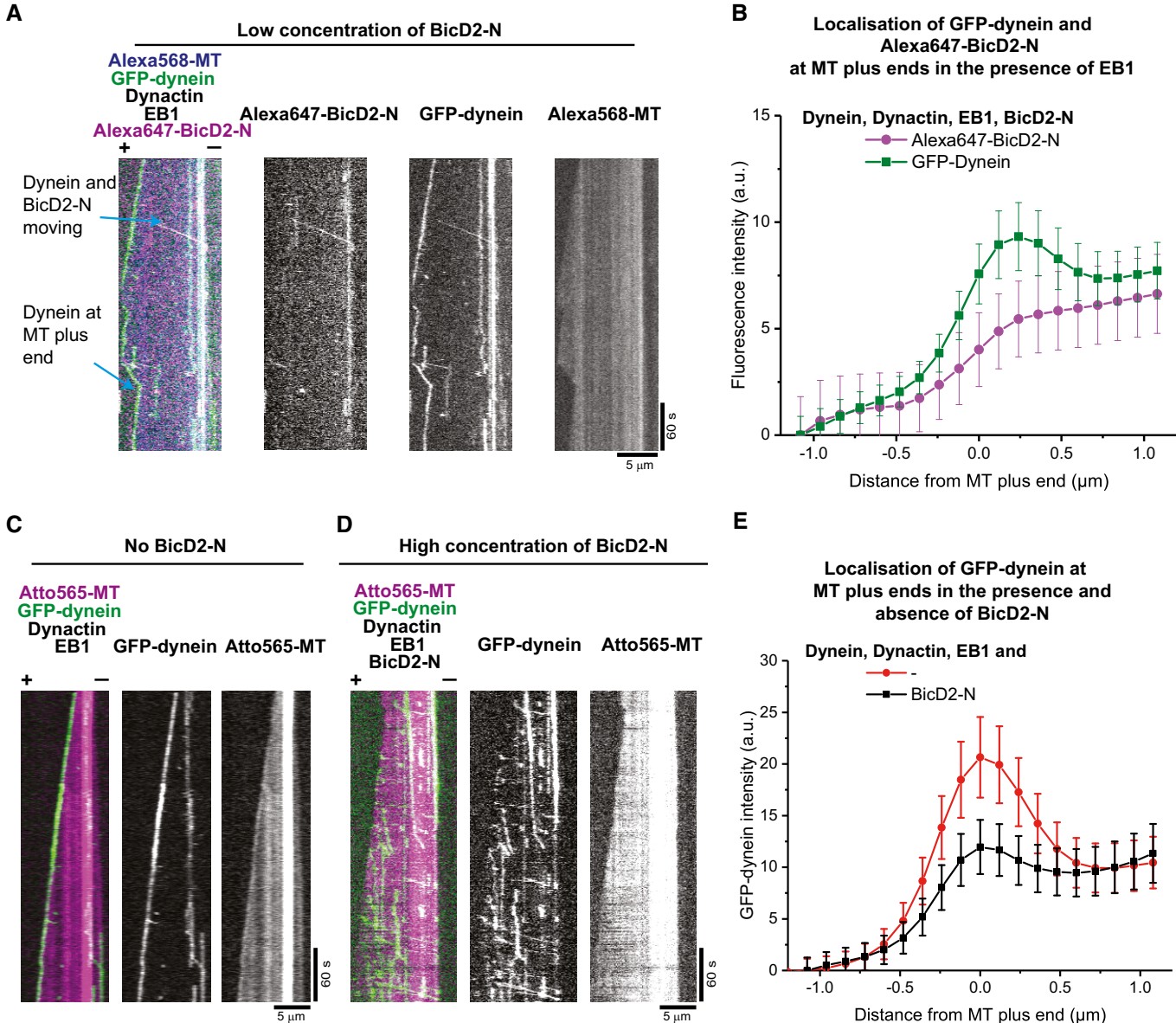

**Figure 2.  Effect of BicD2-N on microtubule end tracking of dynein in the presence of dynactin and EB1.**

A    Merged triple colour and single fluorescence channel TIRF microscopy kymographs of GFP-dynein showing end tracking, processive motility and diffusive and static binding in the presence of all DDB components and EB1. Protein concentrations are 10 nM GFP-dynein (green in merge), 20 nM human dynactin, 20 nM EB1, 200 nM Alexa647-BicD2-N (magenta in merge) and 17.5 μM Alexa568-tubulin (blue in merge). Alexa647-BicD2-N often colocalises with processively moving and occasionally with statically bound GFP-dynein, but not with plus-end tracking GFP-dynein.

B    Averaged fluorescence intensity profiles of GFP-dynein (green squares) and Alexa647-BicD2-N (magenta circles) at growing microtubule plus ends. Mean values from three separate experiments per condition (with a total of at least 75 kymographs) are shown; error bars are s.e.m.

C, D  Dual and single colour kymographs of an Atto565-microtubule (magenta) in the presence of GFP-dynein (green), human dynactin and EB1 and either (C) in the absence or (D) presence of 5 μM BicD2-N; other conditions as in (A). The increased BicD2-N concentration in (D) compared to (A) strongly reduces microtubule plus-end tracking of GFP-dynein.

E    Averaged fluorescence intensity profiles of GFP-dynein at growing microtubule plus ends for the condition without BicD2-N (red circles, as in C) or with 5 μM BicD2-N (black squares, as in D). Mean values from three separate experiments per condition (with a total of at least 106 kymographs) are shown; error bars are s.e.m.

Data information: Microtubule plus and minus ends in merged kymographs are labelled by (+) and (−). Experiments were performed at 30°C.

specific for dynamic microtubule plus ends, as it was not observed for static, GMPCPP-stabilised microtubules (Figs 3D and EV4B). Together, these data show that dynein/dynactin can exist as part of two different populations: either tracking plus ends in an EB-dependent manner or moving processively as part of a DDB complex. EB-dependent end tracking only mildly promotes the initiation of

dynein runs from plus ends, whereas DDB particles have a remarkable inherent preference for starting their runs from growing microtubule plus ends under the experimental conditions used in this study.

When investigating the reason for this bias of motility initiations, we noticed that increasing the dynactin concentration from 20 to

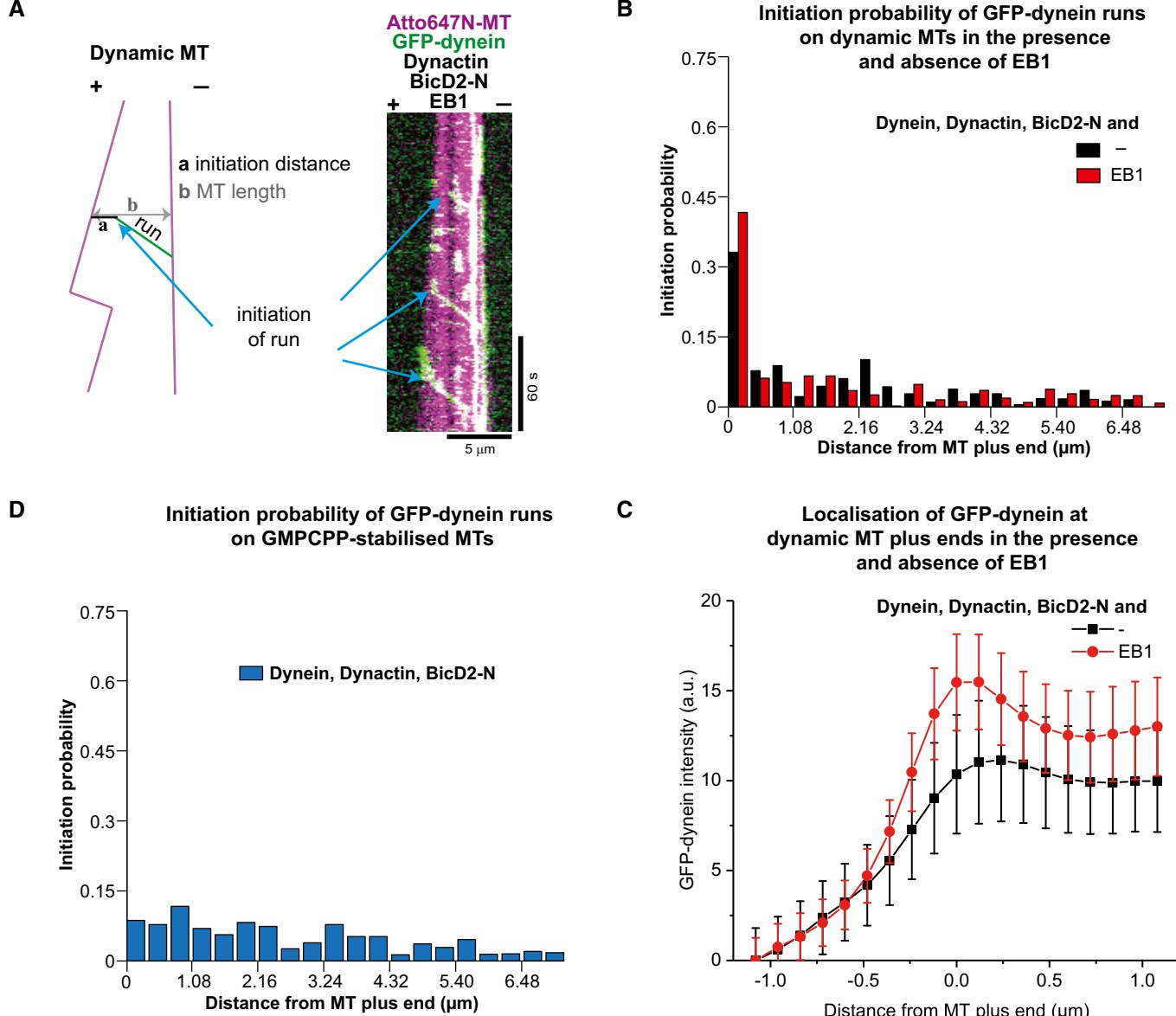

**Figure 3. Spatial initiation probability of processive DDB runs.**

A  Example dual colour kymograph and a schematic showing how the distance of initiation of processive runs was measured. The protein concentrations were 10 nM GFP-dynein, 20 nM pig dynactin, 5 μM BicD2-N, 20 nM EB1 and 17.5 μM Atto647-tubulin.

B  Histograms of the spatial initiation probabilities of processive runs within the first 7.2 μm from growing microtubule plus ends in the absence (black bars), or presence of 10 nM EB1 (red bars); 199 and 221 initiation positions were determined for the two conditions (from three experiments each).

C  Averaged fluorescence intensity profiles of GFP-dynein at growing microtubule plus ends for the conditions as in (B). Mean values from three separate experiments per condition (with a total of at least 69 kymographs) are shown; error bars are s.e.m.

D  Histogram of spatial initiation probabilities of DDB on static GMPCPP microtubules; 294 initiation positions were determined (from three experiments). Concentrations of DDB components as in (B).

Data information: Experiments were performed at 30°C.

80 nM in our dynamic microtubule assay without EB proteins led to some visible plus-end accumulation of GFP-dynein (Fig 4A, Movie EV3), suggesting that dynactin itself might be mediating direct microtubule plus end binding of the DDB complex. To probe this hypothesis further, we purified a fluorescent N-terminal fragment of the p150 subunit of dynactin (p150-N) and observed that 10 nM of p150-N could also recruit GFP-dynein to microtubule plus ends, even in the absence of EB1 (Fig 4B, Movie EV4). Fluorescent p150-N clearly showed preferential localisation to growing microtubule ends, both in the presence (Fig 4B and Movie EV4) and absence (Fig 4C) of dynein. Interestingly, p150-N bound also with high affinity to the stable GMPCPP segments of the microtubules used for surface immobilisation (Fig 4C). These results demonstrate that in the assay buffer used here, the dynactin subunit p150 has an intrinsic, previously unreported preference for growing microtubule end localisation, mediated probably by recognising a conformational aspect of the GTP cap. This end accumulation, however, disappeared at higher ionic strength (Fig 4D), explaining why in previous experiments, which were performed in higher ionic strength buffers, p150-N end accumulation was strictly dependent on EB1 (Duellberg *et al*, 2014), an interaction that appears to be less dependent on the ionic strength.

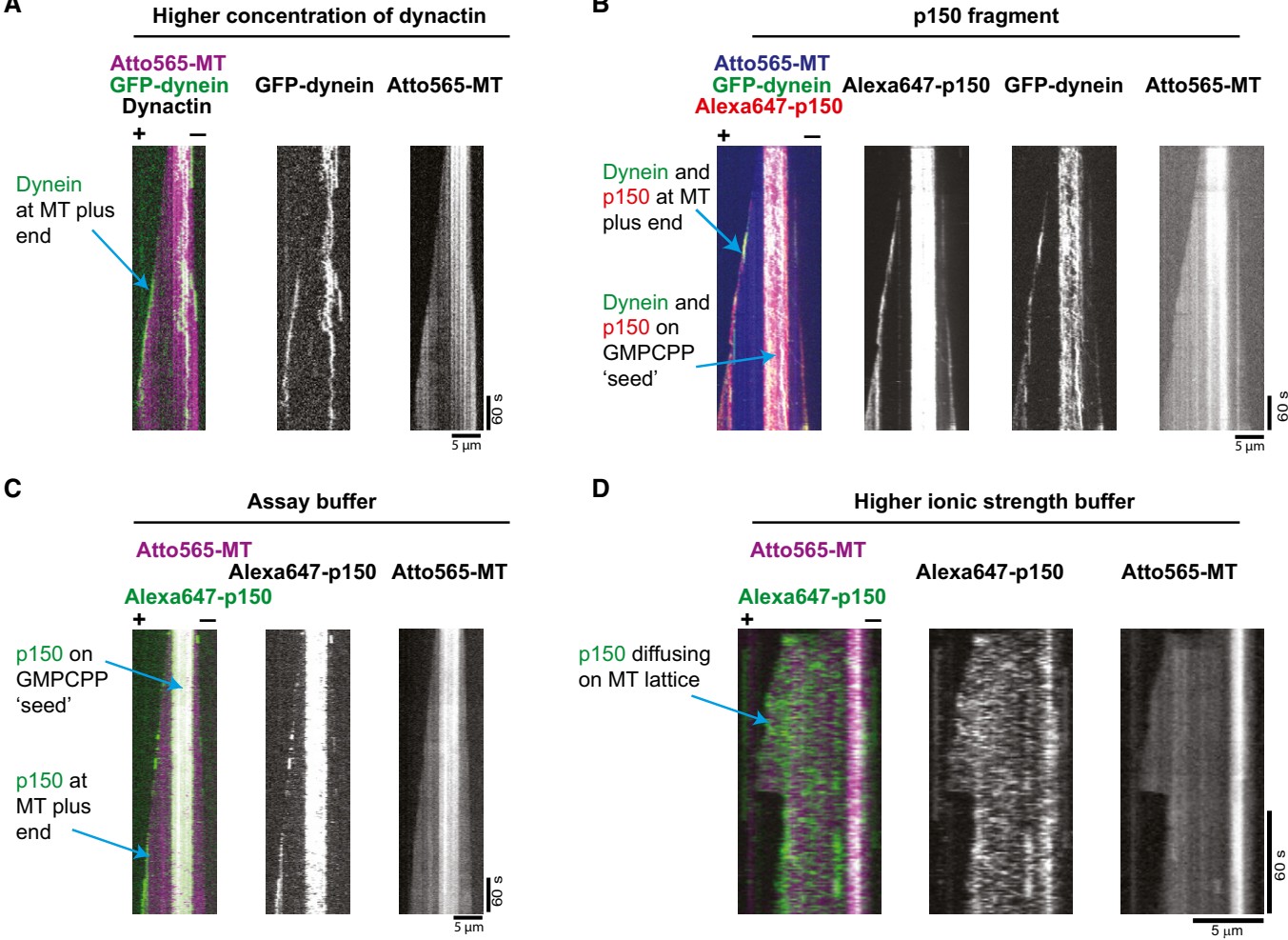

**Figure 4.  p150-dependent plus-end tracking of dynein.**

A, B   Dual and single colour kymographs showing under certain conditions p150 or dynactin-dependent microtubule plus-end tracking of dynein in the absence of EB proteins: (A) 10 nM GFP-dynein (green) tracking the growing end of a Atto565-microtubule (magenta) in the presence of pig dynactin at an elevated concentration of 80 nM. (B) 10 nM GFP-dynein (green) tracking the plus end of an Atto565-microtubule (blue) in the presence of 10 nM Alexa647-p150 (a fragment containing the first 517 amino acids of p150; called p150-N in the text) (red). Experiments in (A) and (B) were performed in standard assay buffer (BRB20 supplemented with 50 mM KCl, for details see Materials and Methods).

C, D   Dual and single colour kymographs showing that microtubule binding behaviour of p150 depends strongly on ionic strength: (C) In standard assay buffer, 10 nM Alexa647-p150 (green) accumulates at the plus end and on the GMPCPP-stabilised segment of a growing Atto565-microtubule (magenta). (D) At higher ionic strength (BRB80 supplemented with 60 mM KCl, see Materials and Methods), Alexa647-p150 (green) binds only weakly to the microtubule without a detectable plus-end preference. Protein concentrations as in (C).

Data information: The Atto565-tubulin concentration was always 17.5 μM tubulin. The temperature was 30°C.

Taken together, our results demonstrate that in addition to the canonical EB1-dependent mechanism, there exists also a so far undescribed mechanism of dynein recruitment to growing microtubule ends depending on direct end recognition by the p150 subunit of the dynactin complex. This interaction explains the bias of the initiation of BicD2-dependent motility from microtubule plus ends, which can be further enhanced by EB proteins.

In metazoan cells, BicD2 and dynactin cooperate with yet another regulator, the Lis1 protein, to control both processive motion as well as end tracking of dynein (Splinter *et al*, 2012). To investigate the mechanism of the combinatorial interplay of these different dynein regulators, we purified a fluorescent Lis1 construct (Fig EV1B) and combined all purified regulators in our dynamic microtubule end tracking assay. Although 5 μM BicD2-N strongly reduced microtubule end tracking of dynein (Fig 5A, and as shown earlier in Fig 2D, Movie EV2), combining 1 μM or 5 μM Lis1 with EB1, GFP-dynein, dynactin and BicD2-N showed that Lis1 can restore end tracking of GFP-dynein in a dose-dependent manner (Fig 5B and C, Movies EV5 and EV6), as quantitatively shown by the GFP-dynein fluorescence profiles at growing microtubule plus ends (Fig 5D). This activity of Lis1 can explain the requirement of Lis1 for end tracking of dynein in cells, where the presence of several cargo adaptors, including BicD2, may suppress EB-dependent end tracking of dynein (Coquelle *et al*, 2002; Splinter *et al*, 2012).

Remarkably, 1 μM of Lis1 also strongly increased the number of processive dynein runs (Fig 5E). However, contrary to the previously reported decrease in the velocity of dynein in gliding assays (Yamada *et al*, 2008; Torisawa *et al*, 2011; Wang *et al*, 2013), we found that neither the velocity nor the run length of the processive DDB complexes moving on dynamic microtubules were affected by the presence of Lis1 (Fig 5F), suggesting that Lis1 might unbind from processive DDB complexes after promoting initiation of motility. To test whether Lis1 does indeed unbind from moving DDB particles after initiation of processive motility, we reduced the concentration of fluorescently labelled Lis1 to decrease the fluorescence background. In triple colour imaging experiments on static microtubules, we observed that only a minority of processive GFP-dynein particles had Lis1 bound (16%, Fig EV5A–D), suggesting that Lis1 can dissociate after motility initiation (see Discussion). Triple colour imaging of Lis1 in the end tracking assay with dynamic

microtubules and all the regulators present at their usual concentrations clearly showed some accumulation of Lis1 at growing microtubule plus ends (Fig EV6A) and again only little colocalisation with processive dynein particles (which in this case may, however, be a consequence of high fluorescence background due to the high Lis1 concentration in solution).

In addition, the initiation probability of processive GFP-dynein runs from the microtubule plus-end region mildly increased in the additional presence of Lis1 (from 0.42 to 0.49, Figs 5G and EV6B). This local effect at growing microtubule ends (comparing the first bins of the histograms shown in Figs 5G and 3B, red) was, however, considerably smaller than the overall effect of Lis1 on initiation of processive motility all along dynamic or static microtubules (Fig 5E or Fig EV5E, respectively).

Together, these data suggest that in the presence of all dynein regulators studied here, Lis1 has a dual role. First, it can act as an initiation factor for processive dynein motility for which BicD2-N and dynactin are additionally required. Lis1 enhances the efficiency of initiation all along the microtubule, only mildly promoting the bias of initiation from the plus-end region. Second, Lis1 can also promote recruitment of dynein/dynactin to growing microtubule plus ends by binding to dynein/dynactin without BicD2-N, thereby apparently depleting the pool of motion-competent DDB particles at high Lis1 concentrations.

# Discussion

In this study, we have dissected the molecular mechanism of how three major dynein regulators control the balance between microtubule end tracking and processive motility of human dynein. Dynactin has a triple role: (i) it mediates EB-dependent plus-end tracking of dynein, in the absence of other regulators (Fig 1), in agreement with recent work (Baumbach *et al*, 2017) (ii) it activates BicD2-dependent dynein processivity (Fig 2), as shown before on static microtubules (McKenney *et al*, 2014; Schlager *et al*, 2014); and (iii) it biases dynein motility initiation from microtubule plus-end regions (Figs 3 and 4). EB-dependent end tracking of dynein/dynactin contributes additionally to biasing motility initiation towards microtubule plus ends, likely by increasing the local concentration of dynein/dynactin there. The microtubule

---

**Figure 5.  Regulation of microtubule end tracking versus processive motility of dynein by Lis1 and BicD2-N in the presence of dynactin and EB1.**

A–C    Kymographs showing GFP-dynein (green) on dynamic Atto647N-microtubules (magenta) in the presence of all the regulators at different Lis1 concentrations. Protein concentrations were 10 nM GFP-dynein, 20 nM pig dynactin, 20 nM EB1, 5 μM BicD2-N and (A) either no Lis1, (B) 1 μM mCherry-Lis1 or (C) 5 μM mCherry-Lis1. Together with BicD2, Lis1 increases the number of processive dynein runs and high concentrations of Lis1 restore plus-end localisation of dynein in the presence of BicD2-N. Microtubule orientation as indicated.

D      Averaged intensity profiles of GFP-dynein at growing microtubule plus ends in the presence of EB1, dynactin and 5 μM BicD2-N (black squares, as in A), in the presence of 1 μM mCherry-Lis1 (blue circles, as in B) and 5 μM mCherry-Lis1 (red triangles, as in C). Mean values from three separate experiments per condition (with a total of at least 49 kymographs) are shown; error bars are s.e.m.

E      The average number of processive DDB runs per μm microtubule length for the three different Lis1 conditions (as in A, B and C). Error bars are s.d. *P = 0.001, **P = 0.003, ***P = 0.001 (unpaired *t*-test); 218, 969 and 646 processive DDB events were analysed for the three conditions, respectively (from two experiments each).

F      Bar graphs showing the mean velocity and the mean run length of processive DDB events for the three Lis1 conditions (as in A, B and C). Error bars are the s.e.m. Mean velocities were 0.37 ± 0.02 μm/s (A), 0.38 ± 0.03 μm/s (B) and 0.35 ± 0.03 μm/s (C). Mean run lengths were 3.1 ± 0.4 μm (A), 3.2 ± 0.5 μm (B) and 2.9 ± 0.3 μm (C). For each condition, over 300 complexes were analysed from three different data sets.

G      Histogram of spatial initiation probabilities of DDB runs in the presence of EB1 and Lis1; 381 initiation positions were determined (from two experiments).

Data information: Experiments were performed at 30°C.

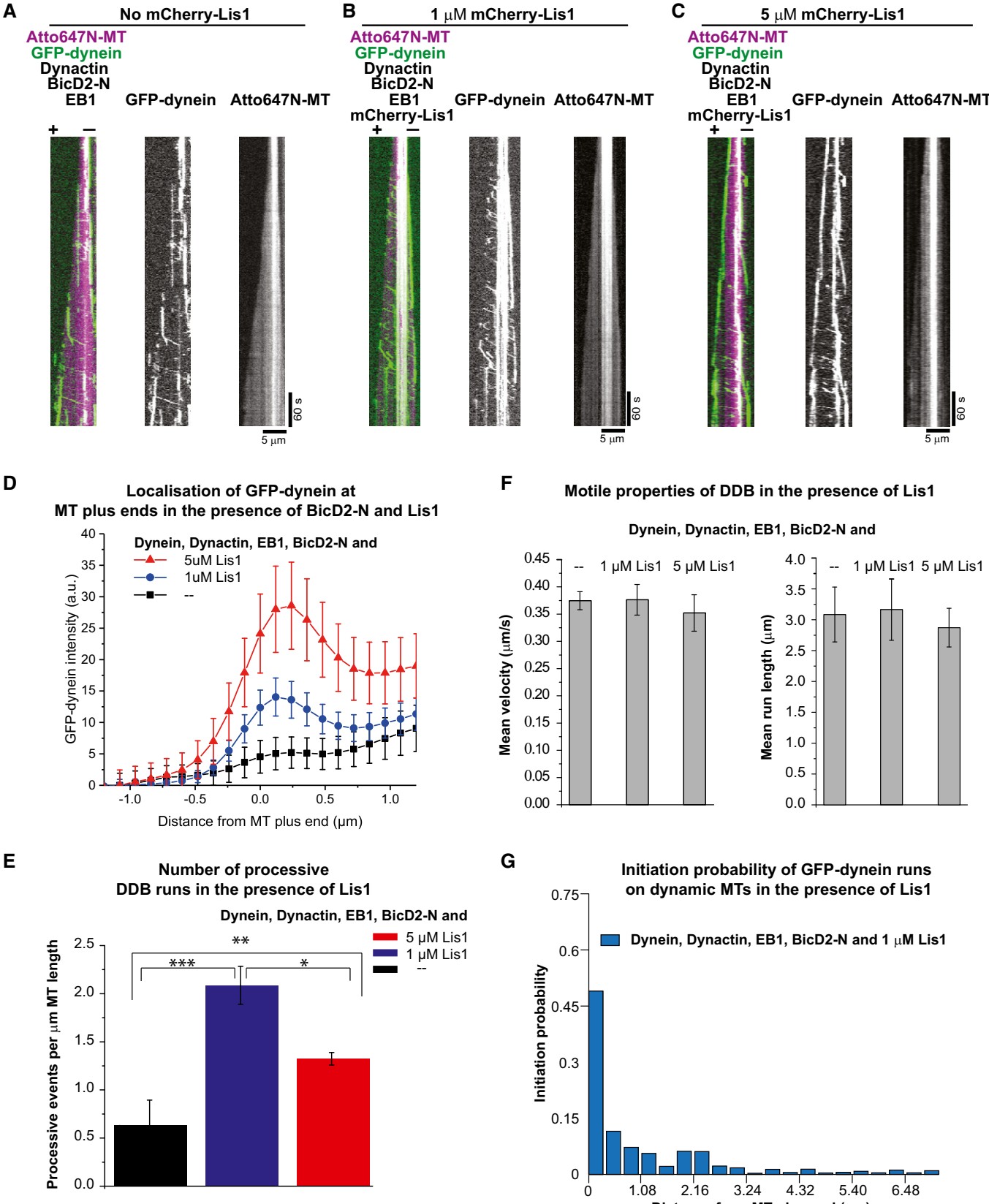

**Figure 5.**

recruitment factor Lis1 adds an additional layer of regulation (Fig 5). It promotes efficient EB-dependent plus-end tracking of dynein/dynactin, especially in the presence of BicD2, and in cooperation with BicD2, Lis1 also promotes initiation of processive motility, in agreement with recent work (Baumbach et al, 2017).

Structural data have suggested a possibility of dynactin adopting an autoinhibited state with its p150 CAP-Gly domain and the first coiled-coil being inaccessible for interaction with EBs and the dynein intermediate chain (Urnavicius et al, 2015). This raised the question of how relevant was the previous reconstitution with an N-terminal fragment of p150 mediating EB1-dependent end tracking of dynein (Duellberg et al, 2014). Here we found that the p150 CAP-Gly domain in the context of the entire dynactin complex is free to associate with EBs in agreement with the variable orientations of the p150 N-terminus observed by cryo-electron microscopy (Urnavicius et al, 2015). The previously noticed weak and likely transient interaction between the dynein and dynactin complexes (McKenney et al, 2014) is apparently sufficient to support EB-dependent microtubule end tracking (Moughamian & Holzbaur, 2012; Moughamian et al, 2013; Duellberg et al, 2014), as our and a recent reconstitution with the entire dynactin complex shows (Baumbach et al, 2017).

In our in vitro reconstitutions, we also observed dynein tracking shrinking microtubule ends, in agreement with recent in vitro studies (Hendricks et al, 2012; Laan et al, 2012; Baumbach et al, 2017) and observations in cells (Ten Hoopen et al, 2012). Future experiments with simultaneously fluorescently labelled individual dynein and dynactin molecules will be required to elucidate the mechanism of shrinking end tracking of dynein and dynactin.

Our combined observations that the dynactin component p150 has the previously undescribed property of biasing processive dynein motility initiation towards microtubule plus-end regions and that EB proteins contribute only mildly to regulate this bias under our conditions predict that processive DDB complexes may not efficiently interact with EB proteins (Fig 6). This would indicate that in the initiating DDB complex, the p150 subunit of dynactin might be conformationally restricted so that its affinity for binding the microtubule surface might be considerably higher than for binding the C-terminus of EB proteins. Alternatively, the EB binding site on microtubules might partially overlap with that of p150 on the microtubule. However, since BicD2-N was not observed to track growing microtubule ends in the presence of end tracking dynein/dynactin, we consider it more likely that the DDB complex is in a conformational state that is less likely to interact with EB1 than with the microtubule directly. This scenario suggests a potentially simple competition-based explanation for why overexpression of the BicD2-N fragment in human cells abolished plus-end tracking of dynein (Splinter et al, 2012). A similar mechanism for down-regulating dynein end tracking by adapter proteins appears to exist in budding yeast where the expression of the cortical adaptor Num1 also removed dynein from microtubule plus ends (Lammers & Markus, 2015).

The intrinsic bias for processive motility initiation of DDB complexes under our conditions may be explained by a selective recognition of the GTP cap region of growing microtubule plus ends by p150, especially since p150 was also observed to bind preferentially to GMPCPP microtubules. This preference for growing microtubule ends and GMPCPP microtubules was, however, not observed

in a recent in vitro study, possibly due to the higher ionic strength buffer used in that study (Baumbach et al, 2017). The nucleotide sensitivity of p150 under our experimental conditions is strikingly similar to that of the unrelated microtubule binding protein TPX2 that was also observed to strongly bind to GMPCPP microtubules and to growing microtubule ends independently of EB proteins (Roostalu et al, 2015). Possibly, this property is related to the reported microtubule nucleation and microtubule catastrophe suppressing activity that p150 and TPX2 have in common (Lazarus et al, 2013; Roostalu et al, 2015). In the future, it will be interesting to study how p150 interacts with the microtubule in its GTP-bound state as this is likely the relevant configuration for transport initiation. It will be interesting to understand how this interaction mode compares to the reported structure of the p150 CAP-Gly domain and its neighbouring acidic region binding in a flexible manner to the C-terminal tubulin tails of GDP/taxol microtubules (Wang et al, 2014).

In cultured mammalian cells, end tracking of dynein was reported to be reduced in the absence of Lis1 (Coquelle et al, 2002; Splinter et al, 2012). This can be explained by our finding that Lis1 appears to compete with the adapter protein BicD2 counteracting the BicD2-mediated reduction in dynein/dynactin end tracking. This suggests that in cells the negative effect of cargo adapters on dynein end tracking would be, at least in part, compensated for by Lis1. This is in agreement with data from budding yeast where overexpression of the Lis1 homologue Pac1 rescued plus-end localisation of dynein in cells overexpressing Num1 adaptor protein (Lammers & Markus, 2015).

Several biochemical and structural data hint at the possibility of an allosteric mechanism underlying the competition between Lis1 and BicD2 for dynein binding. Although both regulators are known to interact with both dynein and dynactin, currently no overlapping binding sites are known. Lis1 binds the AAA+ ring of the dynein motor domain (Tai et al, 2002; Huang et al, 2012) and the p50 subunit of dynactin (Tai et al, 2002), whereas BicD2 interacts with the dynein tail and the Arp1 filament of dynactin (Chowdhury et al, 2015; Urnavicius et al, 2015). Interestingly, Lis1 has been shown to bind to a compact dynein conformation with docked motor domains (also known as the phi particle) (Toba et al, 2015) that has been proposed to correspond to an autoinhibited state of the dynein motor (Torisawa et al, 2014). In contrast, in the DDB particle the motor domains of dynein show a splayed configuration, possibly corresponding to its processive conformation (Urnavicius et al, 2015; Carter et al, 2016). Therefore, it is possible that Lis1 and BicD2 bind preferentially to different dynein conformations, and their competition might shift the equilibrium between the inactive and processive dynein configurations (Fig 6). As such, our data also predict that the EB- and dynactin-dependent plus-end tracking dynein particles may still be in a compact or inhibited state as splaying of dynein would additionally require interaction with BicD2. Such a conformational and hence functional difference between end tracking and processive dynein/dynactin agrees with conclusions from another recent in vitro reconstitution (Baumbach et al, 2017).

Lis1 is well known to be involved in the initiation of dynein-dependent transport in cells (Lenz et al, 2006; Egan et al, 2012; Moughamian et al, 2013). Reducing Lis1 levels in fungi and mammalian cells inhibits dynein-dependent transport of several

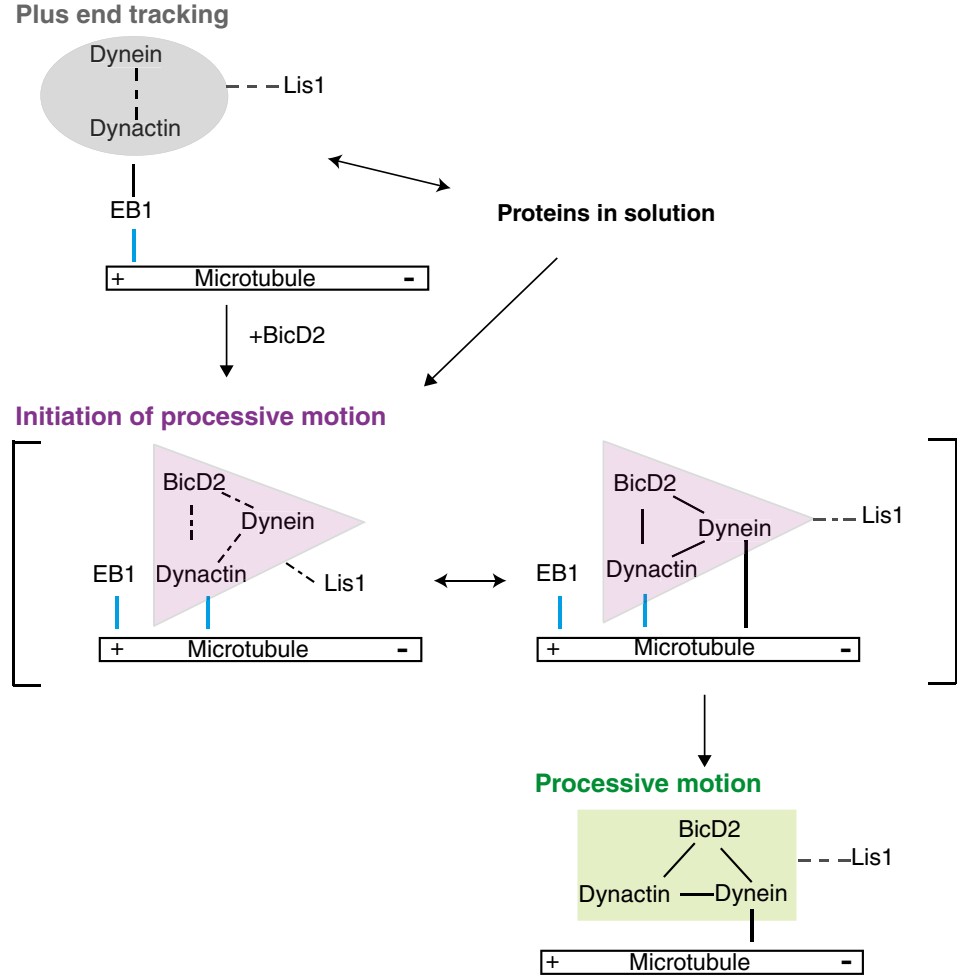

**Figure 6.   Model of the regulation of the balance between dynein plus-end tracking and the initiation of processive minus-end-directed motility.**

Three different hypothetical conformational states of dynein/dynactin characteristic for (i) EB and dynactin co-dependent plus-end tracking, (ii) p150-dependent initiation and (iii) BicD2 and dynactin co-dependent processive motility are indicated by different coloured shaded areas. The dynactin component p150 interacts with the microtubule during initiation in a specific, GTP-sensitive manner, very different from its interaction with EBs during end tracking. Lis1 can stabilise the end tracking complex and in conjunction with BicD2 also promote the formation of the initiation complex, which is incompatible with EB-dependent plus-end tracking. Lis1 may or may not dissociate from the processive dynein/dynactin/BicD2 complex. Blue lines indicate GTP-sensitive interactions with the microtubule.

organelles (Pandey & Smith, 2011; Egan *et al*, 2012; Dix *et al*, 2013; Moughamian *et al*, 2013; Klinman & Holzbaur, 2015) and BicD2-N carrying vesicles (Splinter *et al*, 2012) in agreement with the simultaneous requirement of Lis1 and BicD2 for transport initiation as observed in our minimal system. A positive effect of Lis1 on DDB initiations was also noted in one (Baumbach *et al*, 2017), but not in another (Gutierrez *et al*, 2017) recent *in vitro* study. Also for cells, some opposing findings were reported: a reduction in Lis1 levels in Drosophila cells was shown to enhance dynein-dependent transport of mitochondria (Vagnoni *et al*, 2016), suggesting that Lis1 can also have an inhibitory effect on transport initiation, potentially when it is in excess over cargo adaptors. The contrasting behaviours observed *in vitro* and in cells may be explained by the possible existence of an optimal Lis1/adapter protein ratio for efficient initiation of processive motility, as our data suggest (Fig 5).

Reports of the effect of Lis1 on the velocity of dynein motility vary. *In vitro* experiments with purified proteins showed that Lis1 slowed down dynein motility in gliding assays where teams of surface-immobilised motors propel microtubules (Yamada *et al*, 2008; Torisawa *et al*, 2011; Wang *et al*, 2013; Baumbach *et al*, 2017; Gutierrez *et al*, 2017). The presence of dynactin was reported to revert this slowdown (Wang *et al*, 2013). In filamentous fungi Lis1 (Egan *et al*, 2012) or in *in vitro* single molecule experiments with purified proteins in the absence of mechanical load (McKenney *et al*, 2010), the velocity of dynein was unaffected by the presence of Lis1, similar to our observation of Lis1 not affecting the speed of dynein. The absence of an effect of Lis1 on DDB speed seems to go in line with only a small fraction of DDB having detectably Lis1 bound under our conditions, suggesting that typically Lis1 unbinds after initiation of processive motility. One other recent *in vitro* study of DDB motility in the presence of Lis1 on static microtubules also reported only a mild effect of Lis1 on DDB speed and only partial colocalisation (Gutierrez *et al*, 2017). In contrast, however, strong colocalisation

of processive DDB and Lis1 combined with a marked acceleration of dynein speed by Lis1 *in vitro* was also reported (Baumbach *et al*, 2017). It is currently unclear which conditions cause these contrasting behaviours in *in vitro* motility experiments. It is possible that different binding modes of Lis1 to dynein/dynactin/ adapter protein complexes exist. Under some conditions, Lis1 may release from DDB after initiation of processive motility, whereas under other conditions it may remain bound leading to acceleration of motility (Fig 6).

In conclusion, our study reveals separate roles for each of the three key regulators of dynein activity and shows that dynein can exist in two functionally different pools, either as a non-motile EB-dependent microtubule plus-end tracking complex or as part of a DDB complex with activated processivity. In the DDB complex, the CAP-Gly domain of the dynactin subunit p150 is responsible for biasing initiation of processive motility at microtubule plus ends and on freshly grown tyrosinated microtubules (McKenney *et al*, 2016). Lis1 acts as a general initiation factor that appears to compete in a complex manner with processivity enhancing adapter proteins for dynein and dynactin binding. Biochemical separation of these two dynein pools might allow for independent control of dynein delivery to target structures and of dynein-mediated processive transport.

In the future, structural studies of the dynein complex together with its three major regulators will likely provide important mechanistic insight into the conformations of dynein and dynactin during plus-end tracking versus initiation of transport. In parallel, further extensions of dynamic *in vitro* reconstitutions promise to lead to the dissection and mechanistic understanding of additional layers of regulation of dynein activity.

# Materials and Methods

## Plasmids

To generate a Multibac expression construct of the human cytoplasmic dynein complex labelled with mGFP, we first replaced the SNAP-tag sequence in the pACEBac1 vector (Vijayachandran *et al*, 2013) (generous gift from A. Carter) with the mGFP sequence to generate a plasmid (pGFPdyn1) encoding for a fusion protein consisting of an N-terminal $His_6$-tag followed by a ZZ-tag, a TEV protease cleavage sequence, mGFP and the human cytoplasmic dynein heavy chain ($His_6$-ZZ-TEV-mGFP-DHC). As described (Schlager *et al*, 2014), using purified Cre recombinase (New England Biolabs) this plasmid was combined with pIDC (Vijayachandran *et al*, 2013) (generous gift from A. Carter) that contained the accessory human dynein subunits IC2C, LIC2, Tctex1, LC8 and Robl1 (pdyn2), producing a construct for the simultaneous expression of all six dynein subunits in insect cells. The presence of all subunits in the recombined construct (pGFPdyn3) was confirmed by PCR.

The coding sequence of full-length human Lis1 was amplified from a cDNA (accession number BC064638) by PCR and was cloned into a pFastBac1 plasmid (Invitrogen) to generate a construct for insect cell expression of an N-terminally tagged $His_6$-mCherry-Lis1 where the $His_6$ tag was separated from the mCherry sequence by a TEV protease cleavage site.

The coding sequence for the N-terminal 400 amino acids of human BicD2 was amplified from a cDNA (Origene, SC300552) by PCR and cloned into a pETZT2 plasmid. Sequences were added to generate a bacterial expression construct encoding for a $His_6$-tag followed by a Z-tag, a TEV cleavage site, a SNAP tag, a $Gly_5$ linker and the N-terminal BicD2 fragment ($His_6$-SNAP-BicD2-N in brief).

The coding sequence of full-length human EB3 was amplified by PCR from the plasmid pET28a-His-mCherry-EB3 (generous gift from M. Steinmetz), keeping the "long linker" between mCherry and EB3 (Montenegro Gouveia *et al*, 2010). Sequences were combined in a pETMZ plasmid to generate a bacterial expression construct encoding for a fusion protein consisting of a N-terminal $His_6$-tag followed by a Z-tag, a TEV protease cleavage site, a SNAP tag, the "long linker" and EB3 ($His_6$-SNAP-EB3 in brief).

The sequences of all constructs were verified by sequencing.

## Purification of recombinant human dynein

Human GFP-dynein was expressed in *Sf*21 insect cells and purified by immobilised metal affinity chromatography (IMAC) followed by size exclusion chromatography, as previously described (Trokter *et al*, 2012). All purification steps were carried out at 4°C, and the buffers were degassed and chilled to 4°C. Briefly, a pellet of 800 ml of cell culture was thawed on ice and resuspended in 100 ml lysis buffer (50 mM HEPES pH 7.4, 250 mM K-acetate, 20 mM imidazole 2 mM $MgSO_4$, 0.25 mM EDTA, 10% glycerol (vol/vol), 0.2 mM Mg-ATP, 1 mM 2-mercaptoethanol (ME)) supplemented with protease inhibitors (Complete-EDTA Free, Roche Applied Science). Cells were lysed using a dounce homogeniser (Wheaton). After clarification of the lysate by ultracentrifugation (183,960 × *g*, 45 min, 4°C), the supernatant was loaded onto a 5 ml HiTrap chelating column (GE Healthcare) loaded with $CoCl_2$. The column was then washed with 200 ml lysis buffer. The protein was eluted using elution buffer (50 mM HEPES, pH 7.4, 250 mM K-acetate, 350 mM imidazole, 2 mM $MgSO_4$, 0.25 mM EDTA, 10% glycerol (vol/vol), 0.2 mM Mg-ATP, 1 mM ME). The dynein-containing fractions were pooled and exchanged to gel filtration buffer (30 mM HEPES pH 7.4, 150 mM K-acetate, 2 mM $MgSO_4$, 0.5 mM EGTA, 10% glycerol (vol/vol), 0.05 mM Mg-ATP, 10 mM ME). The $His_6$ tag was cleaved off by incubating with a TEV protease overnight at 4°C. After TEV cleavage, the protein was concentrated to a 5 ml volume using Vivaspin concentrator (Sartorius) with a 50 kDa molecular weight cut-off. The concentrated protein was further purified by size exclusion chromatography using a Superose 6 XK 16/70 prep grade column (GE Healthcare) equilibrated in gel filtration buffer. The mGFP-dynein complex-containing peak fractions were identified by SDS–PAGE Coomassie Blue G-250 staining. The fractions of interest were pooled and concentrated to approximately 0.3 mg/ml using a Vivaspin 50 kDa molecular weight cut-off concentrator and ultracentrifuged (174,000 × *g*, 10 min, 4°C). Protein aliquots were flash-frozen and stored in liquid nitrogen.

## Dynactin purification from HeLa S3 cells

Native human dynactin complex was purified from HeLa S3 cells by BicD2-N affinity chromatography followed by ion exchange chromatography. To generate a BicD2-N column for dynactin affinity purification, 60 mg of purified $His_6$-SNAP-BicD2-N (see below) was

conjugated to a 5 ml NHS column (GE Healthcare) following the conjugation method described by (Widlund *et al*, 2012).

Frozen cell pellets from 25 l HeLa S3 cell culture were thawed and resuspended in approximately 150 ml lysis buffer (30 mM HEPES, pH 7.2, 50 mM K-acetate, 1 mM EGTA, 5 mM Mg-acetate, 0.5 mM ATP, 2 mM ME), supplemented with 10 µg/ml DNAse I, and protease inhibitors (Complete-EDTA Free, Roche Applied Science) using a Polytron tissue dispenser by three pulses of $6.0 \times 10^3$ rpm for 90 sec interspersed by 90-s incubations on ice. The lysate was clarified by centrifugation ($125{,}200 \times g$, 40 min, 4°C). The supernatant was recovered and filtered using a 0.22-µm Steritop filter (Millipore, Bedford, MA), and loaded onto a BicD2 column at 0.5 ml/min. The column was washed with 10 column volumes of wash buffer (lysis buffer supplemented with 100 mM NaCl). The dynein–dynactin complex was then eluted by applying a linear NaCl gradient in lysis buffer using an Akta Purifier FPLC system (GE Healthcare). The dynactin-enriched fractions were identified by SDS–PAGE analysis. Four such BicD2 affinity purifications were run in parallel per day (4 BicD2 columns, 4 × 25 l HeLa S3 cell lysate), for a total of 3 days (300 l HeLa S3 cell lysate in total). The dynactin-containing fractions from all rounds of BicD2 affinity purification were pooled, flash-frozen and stored at −80°C. Between each round of purification, the four BicD2 columns were extensively washed with 1 M NaCl and 500 mM NaCl dissolved in lysis buffer and stored at 4°C. For long-term storage, the BicD2 columns were washed with 100 ml of 6× PBS, then with 100 ml of 1× PBS and finally with 1× PBS with 50% glycerol (vol/vol) for storage at −20°C.

The pooled dynactin-enriched fractions were thawed followed by buffer exchange to MonoQ buffer (35 mM Tris pH 7.2, 5 mM MgSO₄, with 0.5 mM ATP and 2 mM ME) using HiPrep 26/60 desalting columns (GE Healthcare). After buffer exchange, the protein was ultracentrifuged ($174{,}000 \times g$, 20 min, 4°C) and loaded on a MonoQ 5/50 GL column (GE Healthcare) pre-equilibrated with MonoQ buffer. The dynactin complex was eluted by applying a step gradient of NaCl dissolved in MonoQ buffer using the elution regime described by Bingham *et al* (1998). The dynactin complex eluted at 390 mM NaCl as identified by Coomassie Blue-stained SDS–PAGE analysis, and further verified by Western blotting using an anti-p150 antibody (BD-bioscience) and mass spectrometry. The dynactin fractions were pooled and buffer exchanged to storage buffer (30 mM HEPES, pH 7.2, 50 mM K-acetate, 1 mM EGTA, 2 mM Mg-acetate, 0.5 mM ATP, 2 mM ME, 10% glycerol (vol/vol)), concentrated to 0.3 mg/ml using Vivaspin concentrator (Sartorius) with a 50 kDa molecular weight cut-off, ultracentrifuged ($174{,}000 \times g$, 10 min, 4°C), flash-frozen in 5 µl aliquots and stored in liquid nitrogen. Approximately 0.8 mg of human dynactin was obtained from 300 l HeLa S3 cell culture. The final protein concentration was 0.3 mg/ml.

### Dynactin purification from pig brain

Native dynactin was purified from pig brain following the same method as described above. Compared to the large volume of starting material required for purification of human dynactin from HeLa S3, a much smaller volume of material is required when purifying from pig brains, offering a faster alternative for the purification of mammalian dynactin. Compared to the purification from Hela S3 cells, the purification from pig brains differed in the following manner: (i) two frozen pig brains were cut into small pieces with a

scalpel and supplemented with 2 volumes of ice-cold lysis buffer. The brains were then thawed on ice and homogenised using a Polytron dispenser following the same cycle as described for the HeLa S3 cell lysis above. (ii) The homogenate after lysis was centrifuged twice, first: $29{,}000 \times g$, 30 min, at 4°C; and second: $125{,}200 \times g$, 40 min, 4°C. (3) The clarified, unfiltered lysate from two brains was loaded onto one BicD2 column, and the first round of purification was done in 1 day. The BicD2 column eluate was stored at 4°C overnight and loaded onto a MonoQ 5/50 HR column the next day, following the same procedure as described for purification from the Hela S3 cells. Approximately 0.2 mg of dynactin was obtained from two pig brains. The final protein concentration was 0.2 mg/ml. The experiments with Lis1 (Figs 5, EV5, and EV6) were performed using purified pig dynactin.

### Purification and labelling of BicD2-N

His₆-SNAP-BicD2-N was expressed in *Escherichia coli* (BL21 pRIL) by induction with 1 mM IPTG for 16 h at 18°C and purified by IMAC as described below. The pellets from 4 l of cell culture were thawed and resuspended in lysis buffer (50 mM HEPES pH 7.4, 100 mM KCl, 10 mM imidazole, 1 mM ME, 0.1 mM ATP) and supplemented with protease inhibitors (Complete-EDTA Free, Roche Applied Science). The cells were lysed using a microfluidiser. The lysate was clarified by ultracentrifugation ($184{,}000 \times g$, 45 min, 4°C) and loaded onto a 5 ml HisTrap column (GE Healthcare) pre-equilibrated with the lysis buffer. The column was then extensively washed with lysis buffer. The protein was eluted with elution buffer (50 mM HEPES pH 7.4, 100 mM KCl, 300 mM imidazole, 1 mM ME, 0.1 mM ATP). The protein-containing fractions were pooled and dialysed into gel filtration buffer (50 mM HEPES, pH 7.4, 100 mM KCl, 1 mM ME, 0.05 mM ATP, 10% glycerol (vol/vol)). The His₆ tag was then cleaved off by overnight incubation with TEV protease at 4°C. After TEV cleavage, the SNAP-BicD2-N protein was concentrated and further purified by gel filtration using a Superdex 200 10/300 GL column (GE Healthcare) pre-equilibrated with gel filtration buffer. The fractions containing SNAP-BicD2-N (BicD2-N in brief) were pooled, concentrated to 2 mg/ml, ultracentrifuged ($174{,}000 \times g$, 10 min, 4°C) and flash-frozen in 5-µl aliquots in liquid nitrogen.

To generate a fluorescently labelled SNAP-BicD2-N (referred to as Alexa647-BicD2-N), the His₆-SNAP-BicD2-N was incubated overnight at 4°C with an equimolar concentration of SNAP-Surface Alexa Fluor 647 (New England Biolabs) during the TEV protease cleavage reaction. The labelled protein was then passed through HiPrep 26/60 desalting columns (GE Healthcare) to remove the unreacted dye. Size exclusion chromatography and protein flash freezing were performed as described for the unlabelled protein. Final labelling ratio was 0.81 fluorophores per BicD2-N (monomer).

The SNAP-BicD2-N protein that was used for generating the BicD2 column for dynactin purification was dialysed into gel filtration buffer after IMAC elution, ultracentrifuged ($174{,}000 \times g$, 10 min, 4°C), flash-frozen and stored at −80°C. The final protein concentration was here 0.9 mg/ml.

### Purification of EB1 and EB3

Untagged EB1 was expressed and purified as described (Roostalu *et al*, 2015). The final protein concentration was 1.2 mg/ml. To

generate Alexa647-EB3, His$_6$-SNAP-EB3 was expressed in *E. coli* (BL21 pRIL) induced by 0.2 mM IPTG for 16 h at 18°C and purified by IMAC as follows. The pellets were thawed and resuspended in lysis buffer (50 mM NaPi pH 7.2, 400 mM KCl, 5 mM MgCl$_2$, 0.5 mM ME) supplemented with protease inhibitors (Complete-EDTA Free, Roche Applied Science). The lysate was clarified by centrifugation (184,000 × *g*, 45 min, 4°C) and passed over a 5 ml HiTrap Chelating column (GE Healthcare) loaded with CoCl$_2$ pre-equilibrated with lysis buffer. The column was then extensively washed with wash buffer (50 mM NaPi, pH 7.2, 400 mM KCl, 15 mM imidazole, 5 mM MgCl$_2$, 0.5 mM ME). The protein was eluted with elution buffer (50 mM NaPi, pH 7.2, 400 mM KCl, 400 mM imidazole, 5 mM MgCl$_2$, 0.5 mM ME). The protein-containing fractions were pooled, and the buffer was changed to lysis buffer using PD-10 Desalting columns (GE Healthcare). In order to fluorescently label the protein, an equimolar concentration of SNAP-Surface Alexa Fluor 647 (NEB) was added followed by overnight incubation at 4°C in the simultaneous presence of TEV protease cleavage. The unreacted dye was then removed by buffer exchange via HiPrep 26/60 desalting columns (GE Healthcare) followed by size exclusion chromatography using a Superdex 200 10/300 GL column (GE Healthcare). The peak fractions containing Alexa647-labelled SNAP-EB3 were pooled, ultracentrifuged (174,000 × *g*, 10 min, 4°C), aliquoted and flash-frozen in liquid nitrogen. The final protein concentration was 0.6 mg/ml.

### Purification of Lis1 constructs and labelling

His$_6$-mCherry-Lis1 was expressed in Sf21 cells. The cell pellet from a 600 ml culture was resuspended in lysis buffer (50 mM HEPES, pH 7.4, 100 mM KCl, 1 mM ME, 0.05 mM ATP, 10% glycerol (vol/vol)), supplemented with protease inhibitors (Complete-EDTA Free, Roche Applied Science). The cells were lysed using a dounce homogeniser. The lysate was first clarified by centrifugation (184,000 × *g*, 45 min, 4°C) followed by incubation with 1 g of Proteino Ni-TED resin (Macherey-Nagel) in a batch format. The resin was then transferred to a column and extensively washed with lysis buffer. The bound His$_6$-mCherry-Lis1 was cleaved and released from the resin by TEV protease dissolved in lysis buffer incubated for 1 h at room temperature while re-suspending the resin every 5 min. The resin was then separated from the eluted protein by centrifugation (at 1,000 × *g*, 2 min, 4°C). The mCherry-Lis1 protein was further purified by gel filtration using a Superdex 200 10/300 GL column (GE healthcare) pre-equilibrated with lysis buffer. The mCherry-Lis1-containing fractions were identified by SDS–PAGE page, pooled, ultracentrifuged (174,000 × *g*, 10 min, 4°C), supplemented with 20% glycerol (vol/vol), flash-frozen in 5-µl aliquots and stored in liquid nitrogen. The final protein concentration was 2 mg/ml.

### Purification and labelling of p150

His$_6$-SNAP-p150-N (containing the first N-terminal 547 amino acids of the neuronal isoform of p150[Glued]) was purified and labelled with Alexa647 as described (Duellberg *et al*, 2014).

### Tubulin purification and labelling

Porcine brain tubulin was purified as described in Castoldi and Popov (2003) and covalently labelled either with NHS-biotin (Pierce), NHS-Alexa568 (Life technology), NHS-Atto565 (Sigma-Aldrich) or NHS-Atto647N (Sigma-Aldrich) as described by Hyman *et al* (1991).

### Total internal reflection fluorescence microscopy

The flow cells for TIRF microscopy experiments were assembled from a passivated biotin-PEG (polyethylene glycol) functionalised glass coverslips and a poly (L-lysine)-PEG (SuSoS)-passivated counter glass as described previously (Bieling *et al*, 2010). Fluorescently labelled biotinylated GMPCPP-stabilised microtubule "seeds" for TIRF assays with dynamic microtubules were prepared as described previously (Bieling *et al*, 2010) (containing 10% of either Alexa568-, Atto565- or Atto647N-labelled tubulin). The assay was modified from the protocol developed earlier (Bieling *et al*, 2010). Briefly, a flow cell was first incubated for 5 min with 5% Pluronic F-127 (Sigma-Aldrich) at room temperature, washed with BRB80 (80 mM PIPES, 1 mM EGTA, 1 mM MgCl$_2$) supplemented with 50 µg/ml κ-casein (BRB80κ), followed by an incubation with 50 µg/ml of NeutrAvidin in BRB80κ (Life Technologies) on ice for 5 min and washing with BRB80. After warming the flow cell to room temperature, Alexa568-labelled biotinylated GMPCPP "seeds" diluted in BRB80 were passed through and incubated for 5 min for attachment, and then sequentially washed with BRB80, followed by Assay Buffer (AB) (20 mM K-PIPES (pH 6.9), 50 mM KCl, 1 mM EGTA, 2 mM MgCl$_2$, 5 mM ME, 0.1% methylcellulose (4,000 cp, Sigma), 20 mM glucose, 2 mM GTP and 2 mM ATP). The final assay mixture was then passed through the flow cell, which was then sealed using vacuum grease (Beckman), immediately followed by TIRF microscopy imaging.

The final assay mixtures for the different experiments always consisted of (i) a concentrated protein mix of different stored proteins that was first incubated for 5 min (except DDB which was incubated for 10 min) at 4°C, was then diluted by the addition of (ii) 6 µl oxygen scavenger–tubulin mix (5 × 180 µg/ml catalase (Sigma), 5 × 750 µg/ml glucose oxidase (Serva), 5 × 17.5 µM of tubulin, containing 4% of either Alexa568-, Atto565- or Atto647N-tubulin, in BRB80) and was finally brought to 30 µl by adding (iii) the appropriate volume of AB. The final tubulin concentration was always 17.5 µM. For the different experiments, the concentrated protein mixtures and the final protein concentrations were as follows:

#### GFP-dynein/dynactin/BicD2-N (DDB) motility assay on dynamic microtubules

Concentrated protein mix A (or A′) consisted of dynein, human dynactin and BicD2-N (or Alexa647-labelled SNAP-BicD2-N) at a molar ratio of 1:2:20 in 3.6 µl. Final protein concentrations in the assay were 10 nM GFP-dynein, 20 nM dynactin and 200 nM BicD2-N. For control experiments in the absence of dynactin or BicD2-N, their respective storage buffers were added instead of the protein to maintain the same buffer composition.

#### GFP-dynein microtubule end tracking assay

Concentrated protein mix B consisted of GFP-dynein, human dynactin and EB1 (diluted 1:100 in AB) at a molar ratio of 1:2:2 in 4.8 µl. The final protein concentrations in the assay were 10 nM GFP-dynein, 20 nM human dynactin and 20 nM EB1. For triple

colour imaging of microtubule end tracking (Fig 1C), the concentrated protein mix B′ contained Alexa647-labelled SNAP-EB3 instead of untagged EB1 in 3.1 µl, keeping the molar ratios the same and yielding the same protein concentrations in the assay. For control experiments without dynactin, dynactin storage buffer was added instead of dynactin. For experiments in the absence of ATP (Fig 1G), ATP was omitted from AB, and GFP-dynein and human dynactin were exchanged to the AB without ATP.

### Simultaneous GFP-dynein motility and end tracking assay

For triple colour imaging (Fig 2A), concentrated protein mix D contained GFP-dynein, human dynactin, EB1 and Alexa647-BicD2-N at a ratio of 1:2:2:20 in 6 µl. For dual colour imaging (Figs 2D and 5A), concentrated protein mix E contained untagged SNAP-BicD2-N (buffer exchanged to AB) instead of labelled BicD2-N and the protein ratio was changed to 1:2:2:500 in 9.8 µl. Final protein concentrations in the assay were 10 nM GFP-dynein, 20 nM human dynactin, 20 nM EB1, and 200 nM Alexa647-BicD2-N or 5 µM BicD2-N.

For experiments with Lis1 (Figs 5B and C, EV5, and EV6), concentrated protein mix F or F′ contained GFP-dynein, pig dynactin, EB1, BicD2-N (buffer exchanged to AB) and mCherry-Lis1 at a ratio of 1:2:2:500:100 or 1:2:2:500: in 14.9 µl. Final protein concentrations were 10 nM GFP-dynein, 20 nM pig dynactin, 20 nM EB1, 5 µM BicD2-N, 1 µM or 5 µM mCherry-Lis1. For experiments without Lis1 or with lower mCherry-Lis1 concentrations, the buffer composition was kept constant by adding mCherry-Lis storage buffer instead of mCherry-Lis1.

### GFP-dynein motility on GMPCPP microtubules

For motility assays with stabilised microtubules, long GMPCPP microtubules were prepared as described (Roostalu *et al*, 2011) and immobilised in the same manner as described for dynamic microtubule assays. A protein mix consisting of 5 nM GFP-dynein, 10 nM pig dynactin, 200 nM BicD2-N and 50 nM Lis1 was incubated on ice for 10 min and was then added to the appropriate volume of cold assay buffer (AB) or, for control purposes, HEPES-based buffer (60 mM HEPES pH 7.4, 50 mM K-acetate, 2 mM MgCl₂, 10% glycerol (vol/vol), 0.1 mg/ml BSA, 0.5% Pluronic F-127, 10 µM taxol, 0.2 mM κ-casein, 20 mM glucose, 2 mM ATP, 5 mM ME), supplemented with 1.3 mg/ml glucose oxidase (Serva) and 0.66 mg/ml catalase (Sigma). HEPES-based buffer is similar to the buffer used in Gutierrez *et al* (2017). The final volume of the assay mix was 30 µl. This final assay mix was warmed to room temperature and introduced into the flow chamber with immobilised biotin-labelled Alexa647-labelled GMPCPP microtubules.

### TIRF imaging

All samples were imaged at 30°C on a TIRF microscope (iMIC, FEI Munich) described in detail previously (Duellberg *et al*, 2014; Maurer *et al*, 2014; Roostalu *et al*, 2015). Image acquisition and channel alignment were carried out as explained previously (Maurer *et al*, 2014). All time-lapse movies were recorded at 1 fps for 500 s. The exposure time was always 200 ms for all the channels. For dual colour TIRF microscopy imaging, GFP-dynein and Alexa568- or Atto565-labelled microtubules were simultaneously excited at 488 nm and 561 nm, respectively.

For triple colour imaging, either Alexa568-tubulin or mCherry-Lis1 was excited at 561 nm, alternating with simultaneous excitation of GFP-dynein at 488 nm and either Atto647N-tubulin, Alexa647-labelled SNAP-BicD2-N or Alexa647-labelled SNAP-EB3 at 640 nm.

### Analysis of dynein motility

To analyse DDB motion (Figs EV3 and 5F–G), kymographs were generated from image sequences using the KymographClear 2.0 ImageJ plug-in (Mangeol *et al*, 2016) (www.nat.vu.nl/~erwinp/downloads.html). First, a maximum intensity projection image was generated that was used to define tracks using the segmented line tool. After drawing a single track, the plug-in generated three distinct kymographs by Fourier filtering (Mangeol *et al*, 2016) showing separately forward moving, backward moving and pausing, or static particles. The trajectories in the kymographs were further analysed with the KymographDirect software (Mangeol *et al*, 2016). Spurious trajectories were rejected manually and fragmented trajectories from a single track were linked manually using "link" option of the KymographDirect. Trajectories of directionally moving, diffusive or static fluorescent particles are identified by this software using the previously generated Fourier filtered kymographs. Statistical analysis of the velocities and run lengths of the trajectories corresponding to directional motility was then performed automatically by the KymographDirect software.

To calculate the number of directionally motile events per µm of microtubule length (Fig 5E), the total number of directionally motile events for each experimental condition was divided by the total length of dynamic microtubules along which the motile events occurred. The maximum lengths of dynamic microtubules were measured using the segmented line tool in ImageJ. The error bars represent the standard deviation.

### Quantification of fluorescence intensities at microtubule ends

To quantify and compare the fluorescence intensities of GFP-dynein at microtubule plus ends in end tracking experiments, averaged intensity profiles were generated as described previously (Roostalu *et al*, 2015). In brief, kymographs of dynamically growing microtubules were generated using ImageJ Multiple Kymograph plug-in for the GFP-dynein and fluorescently labelled microtubule channels. The growing plus ends were then marked by 3-pixel-wide segmented lines. The kymographs were straightened and aligned using the marked plus end as a reference point. Aligned kymographs were averaged together, and a corresponding image of the standard error of each pixel generated. The average kymograph was then further averaged along the time axis to generate a time-averaged spatial intensity profile for the microtubule plus end. For each assay condition, the GFP-dynein intensity profiles show the mean intensities from at least two pooled data sets. The error bars show the time-averaged standard error produced from the standard error image.

### Quantification of start probability of GFP-dynein run initiation on microtubules

To measure the probability of GFP-dynein run initiation on a dynamic or static microtubule, the distance of the start position of a

run from the microtubule plus end was measured manually from kymographs (initiation distance, Fig 3A). The corresponding microtubule length at the time of initiation was also measured. Probabilities of initiation of DDB runs as a function of the distance from the microtubule plus end were calculated using Matlab. "1 − cumulative probability" distribution functions (1 − cdf) of initiation distances and of the corresponding microtubule lengths at the moment of initiation were computed (Figs EV4 and EV6B). Histograms of relative spatial initiation probabilities within the first 7.2 μm from the microtubule plus end were extracted from the cumulative distributions, correcting for the measured microtubule length distribution (Figs 3B and D, and 5G). A bin size of 360 nm (3 pixels) was chosen, because this is in the range of the length of the EB binding region at growing microtubule ends for the tubulin concentration used here (Bieling *et al*, 2007, 2008; Maurer *et al*, 2011).

Expanded View for this article is available online.

## Acknowledgements

We thank A. Carter for providing the plasmids for dynein Multibac generation, M. Steinmetz for EB3 plasmid and J. Gannon for cloning the EB3 expression construct. We thank the Cell Services Lab for the production of HeLa S3 cells, and the Proteomics Lab for mass spectrometry of dynactin (both at the Francis Crick Institute). This work was supported by the Francis Crick Institute which receives its core funding from Cancer Research UK (FC001163), the UK Medical Research Council (FC001163) and the Wellcome Trust (FC001163). J.R acknowledges funding from a Sir Henry Wellcome Postdoctoral Fellowship (100145/Z/12/Z) and T.S. from the European Research Council (Advanced Grant, project 323042).

## Author contributions

RJ performed the experiments. RJ, NIC and TS analysed data. RJ, JR and TS designed the study and wrote the manuscript. MT cloned Lis1 and p150-N. JR and MT provided biochemistry expertise.

## Conflict of interest

The authors declare that they have no conflict of interest.

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
