## [Review Process File · The EMBO Journal]

Manuscript EMBO-2017-97077

Combinatorial regulation of the balance between dynein microtubule end accumulation and initiation of directed motility

Rupam Jha, Johanna Roostalu, Nicholas, I. Cade, Martina Trokter, Thomas Surrey

Corresponding author: Thomas Surrey, The Francis Crick Institute

Review timeline:

Submission date:	07 April 2017
Editorial Decision:	08 May 2017
Revision received:	12 July 2017
Editorial Decision:	28 August 2017
Revision received:	04 September 2017
Accepted:	13 September 2017

Editor: Ieva Gailite

Transaction Report:

1st Editorial Decision

08 May 2017

Thank you for submitting your manuscript for consideration by the EMBO Journal. We have now received three referee reports on your manuscript, which are included below for your information.

As you can see from the comments, the referees #2 and #3 appreciate that your manuscript adds to our understanding of dynein recruitment regulation. However, the reviewers also raise several substantive concerns with the analysis that would need to be addressed before they can support publication here. From my side, I believe the referee comments to be generally reasonable and addressable. Therefore I would like to invite you to submit a revised version of your manuscript in which you address the comments of all three referees.

When preparing your letter of response to the referees' comments, please bear in mind that this will form part of the Review Process File, and will therefore be available online to the community. For more details on our Transparent Editorial Process, please visit our website: http://emboj.emboPress.org/about#Transparent_Process

We generally allow three months as standard revision time. Please contact us in advance if you would need an additional extension. As a matter of policy, competing manuscripts published during this period will not negatively impact on our assessment of the conceptual advance presented by your study. However, I would appreciate if you would contact me as soon as possible upon publication of any related work in order to discuss how to proceed.

Please feel free to contact me at any time if you have any further questions regarding the revision.

Thank you for the opportunity to consider your work for publication. I look forward to your revision.

REFEREE REPORTS

Referee #1:

Cytoplasmic dynein acts as a motor for the intracellular retrograde motility of vesicles and organelles along microtubules. Dynein associates with dynactin and Bicaudal-D2 (BICD2), which activates its processive motility and is required for the cargo binding. To initiate retrograde transport to the plus-end of microtubules, dynein has to be transported to the plus-end of microtubules, first. However, an underlying mechanism of anterograde transport of dynein and activation of dynein is not yet fully understood. In this work, the authors investigate the transport of dynein and activation using *in vitro* reconstitution experiments using recombinant proteins. Investigation of the dynein transport and activation by its regulators revealed that dynactin and EBs are required for passive dynein transport to the plus-end of microtubules. When N-terminal region of BicD2 is added dynein resumed movements towards the minus-end of microtubules. They further demonstrated that EB independent targeting of dynein to the plus-end of microtubules, suggesting dynein recognize the plus-end of growing microtubules somehow. Finally, they presented LIS1 protein modulates DDB kinetics by potentially dual fashions: enhancement of the initiation of retrograde movement and promotion of dynein/dynactin recruitment to the growing microtubule plus-ends.

While elements of this model might be appealing, a set of disparate experiments is presented. Deeper explorations of the mechanical relevance of regulators are required to be a strong candidate of the EMBO journal.

Major issues.

1. Authors' claim "EB1 family protein (EB)-dependent end tracking constitutes the dominant pathway in non-neuronal cultured mammalian cells" is not correct. Precise characterization has not been performed. Indeed, substantial fraction of vesicles exhibits bidirectional movement, which is frequently described as a "tug-of-war" between oppositely-directed motors attached to the same cargo (Kural et al. 2005, Hendricks et al. 2010). Also, Lis1 disruption displayed aberrant accumulation of dynein around the perinuclear region in fibroblast (Yamada and Toba et al, EMBO 2008). If EB1 family protein (EB)-dependent end tracking constitutes the dominant pathway, available dynein molecules for retrograde transport in non-neuronal cells will be quite limited.
2. Their rate of movement is low for dynein or kinesin-1 (usually >0.5 $\mu\text{m}/\text{sec}$), despite the authors' claims. In addition, authors added recombinant protein simultaneously. However, protein interaction *in vivo* will be carried out by the sequential fashion. For example, the C-terminal region of BicD participates in autoinhibitory interactions with N-terminal CC1 and/or CC2 sequences, thereby attenuating their binding to dynein (Hoogenraad et al. 2001, 2003; Splinter et al. 2012). RAB6 releases this autoinhibition and mediates attachment of Golgi vesicles to the dynein-dynactin complex in Golgi-to-endoplasmic reticulum (ER) transport by BicD (Matanis et al. 2002), thereby ensuring the docking of dynein to neighboring dynactin bound vesicles. Authors have to try to add recombinant proteins sequentially to explore temporal interaction.
3. Authors found that combining 1 mM or 5 mM Lis1 with EB1, GFP-dynein, dynactin and BicD2-N showed that Lis1 can restore end tracking of GFP-dynein in a dose dependent manner. On the other hand, they were also unable to visualise fluorescently labelled Lis1 co-migrating with processive GFP-dynein in triple colour imaging experiments. They proposed two possibilities: Lis1 either did not bind processive DDB complexes or, alternatively, Lis1 does not affect their velocity and run length. It is possible to address this question by simple gilding assay *in vitro*. Also, this reviewer welcomes more persuasive discussion.
4. This reviewer has a serious concern regarding reuse of the data. Figure 3A is quite similar with Figure 5A. I suppose this case is not intentional, rather a careless mistake. However, this issue should be carefully avoided.

Minor issues:

1. Front page. "Lisencephaly-1" should read Lissencephaly-1.
2. It is better to combine Figure 2 and Figure 3. In particular Figure 3B and 3D.
3. Figure 4B: why two "MT plus end" are present in the same figure?

Referee #2:

The manuscript by Jha et al shows in vitro reconstitution of the motion of dynein complexes on dynamic microtubules. There are some new findings in this paper, for example the full dynactin complex seems to be sufficient for recruiting dynein to MT plus ends (EB1 not necessary). However, this work continues on the line of (now too many !!) studies on the in vitro reconstitution of dynein motility on dynamic MTs. There are many components (EB1, dynein, dynactin, Lis1, BicD ...), so many combinatorial possibilities exist for experiments. But, how long does one really want to continue doing this?

That being said, I am a bit on the positive side because these findings may be useful for understanding cellular function of dynein. The authors should address the following points:

- 1) In the absence of a good schematic, the paper is difficult to read and you lose interest at some point. This was mentioned by the Elife referees also, but the authors have not put in such a schematic. This is absolutely required. I suggest a cartoon which shows the new findings and how they compare with what was already known
- 2) Most of the figures just show a kymograph of a microtubule with various dynein/EB1/etc components added. However, there is no sense of how repeatable these experiments are. How many dynamic MTs were observed? How many times was the experiment repeated etc etc?? Were multiple preparations done? I find it hard to decide about this manuscript in the absence of this information.
- 3) Fig 4D and End of page 6 -- dynactin does not bind to MT ends at higher ionic strength. How do these strengths compare with cytosol ?? This has to be described in the paper.

Referee #3:

This paper examines the interplay between dynein regulators (specifically Lis1, dynactin, BicD, and EB1) on dynein plus end tracking and minus end directed motility. The p150 subunit of dynactin has previously been shown to be recruited to MT plus ends via an interaction with the protein EB1. This in turn is sufficient to recruit dynein to the dynamic ends via an interaction with its intermediate chain. In this paper, the authors expand upon this work from 2014 by examining the effect of the entire dynactin complex on dynein plus-end recruitment. In particular, they emphasize the finding that the CAP-Gly domain of p150 is in an active form in the context of the whole dynactin even in the absence of other regulators. The authors are impressively able to reconstitute DDB motility on dynamic microtubules in the presence of EB1/3. These experiments reveal novel insights into dynein recruitment and motility on a microtubule lattice. They find that the dynein population tracking plus ends seems distinct from those moving towards the minus end a DDB complex. High concentrations of the adaptor Bicaudal D2 reduces the number of dyneins tracking on the plus ends. This is restored upon the addition of another dynein regulator, LIS1. In addition to rescuing the end tracking of dynein, LIS1 also appeared to facilitate processive DDB runs, suggesting that it might have multiple functions. Overall, this study is of high quality and sufficient in interest for publication in EMBO J. Some comments are described below.

- 1) The authors mention that neither velocity or processivity of DDB are affected in the presence of LIS1, contrary to reports that LIS1 decreases gliding velocity. They also show that LIS1 does not colocalize with most of their DDB particles in their system. These two observations seem contradictory. Some clarification is needed, especially if Lis1 is not bound to moving dyneins/

2) The authors observed that the initiation frequency is increased in the presence of LIS1 (Fig. 6EV). This can potentially explain the increase in processive DDB runs in Fig. 5E in the 1 μ M condition. Is the prediction that this increase in processive runs would not be observed on static GMPCPP microtubules?

3) Regarding the increase in initiation frequency, is Lis1 actually doing something to dynein to increase initiation (e.g. allosterically) or is it just due to the increased binding at the binding at the plus end leads to proportionally more runs? Perhaps I missed this, but if it just increases plus end binding, it should not be labeled as "increasing initiation". This is an important point that should be clarified.

4) Is the ability of dynein tracking shrinking microtubules dependent on the presence of dynactin?
5) Given that it has previously been shown that LIS1 cannot bind to growing MT ends on its own, is there a difference in the initiation probability when LIS1 is added in the presence vs absence of an EB protein (Fig. EV6)?

6) The 1-cdf graphs in Fig. EV4 for the presence or absence of EB1 seems to show that there is some effect of EB1 on the initiation probability, no? Is saying that the plus end initiation bias is "largely unaffected" by EB1 accurate?

7) Recent reports from another laboratory report LIS1/DDB complexes moving on the MT lattice. This paper does not see very many instances of LIS1 co-migrating with a moving DDB. (Fig. EV5). They call LIS1 colocalizations "rare events." Is there a possible explanation, perhaps differences in buffer conditions?

8) A recent paper from Bullock lab doesn't see a difference in the ratio of DDBs initiating from the plus ends/number initiating from the lattice with the addition of LIS1. The data in this paper suggests that LIS1 increases this ratio. Perhaps this difference should be discussed.

1st Revision - authors' response

12 July 2017

Referee #1:

Cytoplasmic dynein acts as a motor for the intracellular retrograde motility of vesicles and organelles along microtubules. Dynein associates with dynactin and Bicaudal-D2 (BICD2), which activates its processive motility and is required for the cargo binding. To initiate retrograde transport to the plus-end of microtubules, dynein has to be transported to the plus-end of microtubules, first. However, an underlying mechanism of anterograde transport of dynein and activation of dynein is not yet fully understood. In this work, the authors investigate the transport of dynein and activation using *in vitro* reconstitution experiments using recombinant proteins. Investigation of the dynein transport and activation by its regulators revealed that dynactin and EBs are required for passive dynein transport to the plus-end of microtubules. When N-terminal region of BicD2 is added dynein resumed movements towards the minus-end of microtubules. They further demonstrated that EB independent targeting of dynein to the plus-end of microtubules, suggesting dynein recognize the plus-end of growing microtubules somehow. Finally, they presented LIS1 protein modulates DDB kinetics by potentially dual fashions: enhancement of the initiation of retrograde movement and promotion of dynein/dynactin recruitment to the growing microtubule plus-ends.

While elements of this model might be appealing, a set of disparate experiments is presented. Deeper explorations of the mechanical relevance of regulators are required to be a strong candidate of the EMBO journal.

Major issues.

1. Authors' claim "EB1 family protein (EB)-dependent end tracking constitutes the dominant pathway in non-neuronal cultured mammalian cells" is not correct. Precise characterization has not been performed. Indeed, substantial fraction of vesicles exhibits bidirectional movement, which is frequently described as a "tug-of-war" between oppositely-directed motors attached to the same cargo (Kural et al. 2005, Hendricks et al. 2010). Also, Lis1 disruption displayed aberrant

accumulation of dynein around the perinuclear region in fibroblast (Yamada and Toba et al, EMBO 2008). If EB1 family protein (EB)-dependent end tracking constitutes the dominant pathway, available dynein molecules for retrograde transport in non-neuronal cells will be quite limited.

Authors: We did not intend to claim that dynein plus end accumulation is overall the dominant behaviour of dynein in non-neuronal metazoan cells. In the Introduction, we mention also dynein's role in bidirectional transport. However, in this study we focus on the mechanism of plus end accumulation of dynein, for which the current state of the literature provides observations of mostly kinesin-dependent end accumulation of dynein in fungi and neurons, and of mostly EB-dependent end accumulation in non-neuronal metazoan cells. We agree that this is still an area under active investigation, as this reviewer points out. We have modified our sentence, now not making a statement about the relative importance of different mechanism (end of page 2 / beginning of page 3).

2. Their rate of movement is low for dynein or kinesin-1 (usually >0.5 $\mu\text{m}/\text{sec}$), despite the authors' claims. In addition, authors added recombinant protein simultaneously. However, protein interaction *in vivo* will be carried out by the sequential fashion. For example, the C-terminal region of BicD participates in autoinhibitory interactions with N-terminal CC1 and/or CC2 sequences, thereby attenuating their binding to dynein (Hoogenraad et al. 2001, 2003; Splinter et al. 2012). RAB6 releases this autoinhibition and mediates attachment of Golgi vesicles to the dynein-dynactin complex in Golgi-to-endoplasmic reticulum (ER) transport by BicD (Matanis et al. 2002), thereby ensuring the docking of dynein to neighboring dynactin bound vesicles. Authors have to try to add recombinant proteins sequentially to explore temporal interaction.

Authors: We agree that speeds of dynein-driven cargoes measured in living mammalian cells have often been reported to be around 1 $\mu\text{m}/\text{sec}$. This is indeed faster than speeds measured for the movement of single purified mammalian dynein/dynactin/BicD2 complexes *in vitro* where ~ 0.5 $\mu\text{m}/\text{sec}$ have been measured (McKenney et al., Science, 2014; Schlager et al., EMBO J., 2014). This difference between single purified complexes and dynein cargoes in cells may be due to differences in temperature, buffer/cytosol composition or multiple dyneins transporting cargoes in cells versus single complexes being observed *in vitro* etc. Here we simply claim that the properties of our dynein/dynactin/BicD2 (DDB) complex are similar to those of complexes previously prepared from purified proteins by other labs (McKenney et al., Science, 2014; Schlager et al., EMBO J., 2014). Regarding the second point of the reviewer: It is useful to note that our experiments are performed with a N-terminal BicD2 fragment that lacks the ability for autoinhibition. We state this now explicitly in the text on page 5 when we introduce the fragment for the first time in the Results section. Therefore, we do not see the necessity for sequential protein additions.

3. Authors found that combining 1 mM or 5 mM Lis1 with EB1, GFP-dynein, dynactin and BicD2-N showed that Lis1 can restore end tracking of GFP-dynein in a dose dependent manner. On the other hand, they were also unable to visualise fluorescently labelled Lis1 co-migrating with processive GFP-dynein in triple colour imaging experiments. They proposed two possibilities: Lis1 either did not bind processive DDB complexes or, alternatively, Lis1 does not affect their velocity and run length. It is possible to address this question by simple gilding assay *in vitro*. Also, this reviewer welcomes more persuasive discussion.

Authors: It is now well established that ensemble dynein behaviour in a gliding assay can differ from single DDB behaviour in the presence of Lis1. Please see the very recent papers that came out after submission of our manuscript: Baumbach, eLife, 2017; Gutierrez et al., JCB, 2017. The origin for these different behaviours in different assays is currently not understood and not subject of our study.

To address this reviewer's concern concerning Lis1-DDB colocalisation, we performed new single DDB motility assays at lower Lis1 concentrations (50 nM) to reduce the Lis1 fluorescence background in order to allow better visualisation of potential Lis1-DDB binding events. Under these conditions, we observed that only 9% of processive dynein complexes had Lis1 bound. This agrees with the recent paper by Gutierrez et al. They reported a slightly higher fraction (17.5%) of DDB-Lis1 particles. In new control experiments, we have reproduced their results using their buffer conditions (we find 16% colocalisation). In agreement with our work, they also report that Lis1 binding has only a minor effect on the speed of processive DDB motion. We have replaced the previous data (in the previous Fig. EV6) by these new data (now in the new Figure EV5). In

contrast, the other recent paper by Baumbach et al. reported strong colocalisation of Lis1 with DDB particles and an acceleration of dynein speed. We have expanded the discussion regarding these interesting similarities and discrepancies between these recent observations (third-last paragraph of the Discussion on page 11).

4. This reviewer has a serious concern regarding reuse of the data. Figure 3A is quite similar with Figure 5A. I suppose this case is not intentional, rather a careless mistake. However, this issue should be carefully avoided.

Authors: We thank the reviewer for pointing this out. We have selected another kymograph for Figure 5A from another dataset recorded under the same conditions.

Minor issues:

1. Front page. "Lisencephaly-1" should read Lissencephaly-1.

Authors: We have corrected the spelling in the abstract.

2. It is better to combine Figure 2 and Figure 3. In particular Figure 3B and 3D.

Authors: We think that this would produce a figure that is too dense and prefer to leave the presentation of these data as they are.

3. Figure 4B: why two "MT plus end" are present in the same figure?

Authors: We have corrected this mistake. One 'MT plus end' label has been replaced by a 'GMPCPP seed' label.

Referee #2:

The manuscript by Jha et al shows in vitro reconstitution of the motion of dynein complexes on dynamic microtubules. There are some new findings in this paper, for example the full dynactin complex seems to be sufficient for recruiting dynein to MT plus ends (EB1 not necessary). However, this work continues on the line of (now too many !!) studies on the in vitro reconstitution of dynein motility on dynamic MTs. There are many components (EB1, dynein, dynactin, Lis1, BicD ...), so many combinatorial possibilities exist for experiments. But, how long does one really want to continue doing this?

That being said, I am a bit on the positive side because these findings may be useful for understanding cellular function of dynein. The authors should address the following points:-

1) In the absence of a good schematic, the paper is difficult to read and you lose interest at some point. This was mentioned by the Elife referees also, but the authors have not put in such a schematic. This is absolutely required. I suggest a cartoon which shows the new findings and how they compare with what was already known

Authors: We have added a conceptual schematic summarising our model of the mechanism regulating the balance between plus-end tracking and initiation of processive motility of dynein/dynactin (new Figure 6).

2) Most of the figures just show a kymograph of a microtubule with various dynein/EB1/etc components added. However, there is no sense of how repeatable these experiments are. How many dynamic MTs were observed? How many times was the experiment repeated etc etc?? Were multiple preparations done? I find it hard to decide about this manuscript in the absence of this information.

Authors: These numbers are stated in the figure legends.

3) Fig 4D and End of page 6 -- dynactin does not bind to MT ends at higher ionic strength. How do these strengths compare with cytosol ?? This has to be described in the paper.

Authors: The osmolarity of cytosol is typically considered to be 300 mM. Part of the solutes are charged and hence contribute to the ionic strength. BRB80 or BRB80 with some tens of mM KCl are often considered similar to the ionic strength of cytosol (e.g. Patel et al., Science, 2017). A more precise answer cannot be given, since proteins also considerably contribute to the ionic strength of cytosol. Our buffer is within the range of buffers that are considered close to physiological in ionic strength. Importantly our buffer allows both microtubules to display normal dynamic instability and also dynein to display normal motility *in vitro*, which is an important experimental constraint. We state this explicitly in the beginning of our Results section.

Referee #3:

This paper examines the interplay between dynein regulators (specifically Lis1, dynactin, BicD, and EB1) on dynein plus end tracking and minus end directed motility. The p150 subunit of dynactin has previously been shown to be recruited to MT plus ends via an interaction with the protein EB1. This in turn is sufficient to recruit dynein to the dynamic ends via an interaction with its intermediate chain. In this paper, the authors expand upon this work from 2014 by examining the effect of the entire dynactin complex on dynein plus-end recruitment. In particular, they emphasize the finding that the CAP-Gly domain of p150 is in an active form in the context of the whole dynactin even in the absence of other regulators. The authors are impressively able to reconstitute DDB motility on dynamic microtubules in the presence of EB1/3. These experiments reveal novel insights into dynein recruitment and motility on a microtubule lattice. They find that the dynein population tracking plus ends seems distinct from those moving towards the minus end a DDB complex. High concentrations of the adaptor Bicaudal D2 reduces the number of dyneins tracking on the plus ends. This is restored upon the addition of another dynein regulator, LIS1. In addition to rescuing the end tracking of dynein, LIS1 also appeared to facilitate processive DDB runs, suggesting that it might have multiple functions. Overall, this study is of high quality and sufficient in interest for publication in EMBO J.

Some comments are described below.

1) The authors mention that neither velocity or processivity of DDB are affected in the presence of LIS1, contrary to reports that LIS1 decreases gliding velocity. They also show that LIS1 does not colocalize with most of their DDB particles in their system. These two observations seem contradictory. Some clarification is needed, especially if Lis1 is not bound to moving dyneins/

Authors: It is becoming clear that the effect of Lis1 on microtubule transport by immobilised dynein in gliding assays differs from the effect of Lis1 on single freely moving DDB complexes on immobilised microtubules (Baumbach et al., eLife, 2017; Gutierrez et al., JBC, 2017). We have confirmed that our purified Lis1 also slows down dynein gliding (not shown), as reported in these publications. This is an interesting area for future investigation. We cite now these results reported in these two very recent papers in the Discussion of our revised manuscript.

2) The authors observed that the initiation frequency is increased in the presence of LIS1 (Fig. 6EV). This can potentially explain the increase in processive DDB runs in Fig. 5E in the 1 μ M condition. Is the prediction that that this increase in processive runs would not be observed on static GMPCPP microtubules?

Authors: The relative initiation probability at the growing microtubule plus end region is increased by Lis1 by 17% (compare Figs. 3B and EV6B). The total number of processive events per microtubule length increases however by ~ 3fold (from 0.62 to 2.08/ μ m) comparing the same conditions (Fig. 5E). This indicates that the overall increase in the number of events does not result mainly from increasing the number of initiations in the growing end region, but that Lis1 has a general stimulating effect all along the microtubule. Therefore the prediction is that Lis1 increases also the number of initiations on static GMPCPP microtubules. We performed new experiments to test this prediction and we indeed observed a general stimulation of processive motility initiation of DDB complexes by Lis1 also on static microtubules. These new data are shown in Fig. EV5E.

3) Regarding the increase in initiation frequency, is Lis1 actually doing something to dynein to increase initiation (e.g. allosterically) or is it just due to the increased binding at the binding at the plus end leads to proportionally more runs? Perhaps I missed this, but if it just increases plus end binding, it should not be labeled as "increasing initiation". This is an important point that should be clarified.

Authors: We agree that this is an important point that needed clarification. As explained in our response to the previous question, the Lis1 effect on initiations is general and not only specific to initiations from the end. Our new data on the overall number of initiations on GMPCPP-stabilised microtubules (Fig. EV5E) (see previous response) make this more clear now. We have added an explicit statement to our Discussion.

4) Is the ability of dynein tracking shrinking microtubules dependent on the presence of dynactin?

Authors: This is difficult to answer, because there is hardly any dynein at growing plus ends in the absence of dynactin. Furthermore, we believe that for a definitive answer to this question, purified fluorescently labelled dynactin complex is required, which we do not have. Therefore, a definitive answer here is unfortunately beyond scope. This does, however, not affect our main conclusions. We prefer to simply report the observation, because it is evident in the data.

5) Given that it has previously been shown that LIS1 cannot bind to growing MT ends on its own, is there a difference in the initiation probability when LIS1 is added in the presence vs absence of an EB protein (Fig. EV6)?

Authors: This concern addresses the same point raised in concerns 2 and 3. As we replied there, we have now clarified that our data indicate that the Lis1 effect is general and not end-specific.

6) The 1-cdf graphs in Fig. EV4 for the presence or absence of EB1 seems to show that there is some effect of EB1 on the initiation probability, no? Is saying that the plus end initiation bias is "largely unaffected" by EB1 accurate?

Authors: We thank this reviewer for pointing this out. The cumulative distributions which are independent of the choice of bin size indeed seem to show an effect of EB1. This was also visible as a very small effect in the histograms of the previously submitted version of the manuscript (with 1 μm bins). We have now recalculated all histograms of the initiation probabilities for smaller bin sizes to visualise the differences in initiation probabilities at higher spatial resolution. We chose 360 nm as a new bin size (~3x smaller than before), because it corresponds to 3 pixels which seems reasonable for our manual determination of the start position of processive runs and which also corresponds roughly to the size of the EB binding region (Bieling et al., Nature, 2007; Maurer et al, PNAS, 2011). The new histograms (Figure 3B) indeed confirm, as suggested by the reviewer, that EB1 has a measurable effect on the start probability within the first 360 nm of the microtubule under our conditions: it increases this local probability by 27% compared to the condition with DDB alone. This is less than the dynactin-mediated effect at a dynamic growing end versus a static end (>250%) and a little more than the additional effect of Lis1 (17%) (Figure 3B, 3D, and 5G). We have rephrased our statements accordingly, both in the Results section (page 6) and in the Discussion (page 8, 9).

7) Recent reports from another laboratory report LIS1/DDB complexes moving on the MT lattice. This paper does not see very many instances of LIS1 co-migrating with a moving DDB. (Fig. EV5). They call LIS1 colocalizations "rare events." Is there a possible explanation, perhaps differences in buffer conditions?

Authors: Since submission of our manuscript, two papers were published where the authors image DDB motion with and without Lis1 in vitro: Baumbach et al., eLife, 2017 and Gutierrez et al., JBC, 2017. Baumbach et al. report 75% colocalisation of Lis1 with DDB and Gutierrez et al report 17.5% colocalisation. Both studies use different buffers, protein concentrations and Lis1 constructs. The buffer in the Baumbach study is PIPES based, like ours and the buffer in the Gutierrez study is HEPES-based. We have made new control experiments using conditions very similar to the Gutierrez study and find that only the minority (16%) of processive DDB complexes has Lis1 bound (Figure EV5 A-D), very similar to their observation, showing that it is independent of the chosen

buffer. On page 11 in the revised manuscript, we discuss discrepancies and similarities of the three studies, noting that in the light of the currently reported observations some aspects of the effect of Lis1 on DDB motility deserve further investigation in the future.

8) A recent paper from Bullock lab doesn't see a difference in the ratio of DDBs initiating from the plus ends/number initiating from the lattice with the addition of LIS1. The data in this paper suggests that LIS1 increases this ratio. Perhaps this difference should be discussed.

Authors: We discuss this difference in the revised version of our manuscript. It is important to note that we define 'plus end region' as the first 360 nm (3 pixels) of the microtubule measured from the plus end which corresponds roughly to the length of the EB binding region. Baumbach et al. uses a different definition for the microtubule end that has subpixel dimensions. Different definitions of the end region or other differing assay conditions may lead to differences in the detectable initiation probabilities. It is worth noting, that our data show, partly in agreement with the Baumbach et al., that the effect of Lis1 on 'end initiation' is much smaller than its overall effect of initiation measured all along the microtubule. Therefore, we do not consider this a major difference.

2nd Editorial Decision

28 August 2017

Thank you for submitting a revised version of your manuscript. I apologise for the delay in communicating the decision due to delayed referee reports. The manuscript has now been seen by one of the original referees, who finds that the main concerns have been addressed. There remain only a few minor editorial issues that have to be resolved before formal acceptance of the manuscript.

REFEREE REPORTS

Referee #3:

Jha et al. have done a reasonable job in addressing. The clarification of whether Lis1 is merely increasing plus end recruitment or imposing a more global lattice effect was important, and I think the GMPCPP experiment in the new Fig. EV5E as well as a more explicitly stated sentence in the discussion helps to answer this question. The experiment examining LIS1 localization in a different buffer condition was informative and reduces the likelihood that the difference in co-localization between this study and that of Baumbach et al. is due to a simple switch between PIPES and HEPES. While this still leaves the question as to why LIS1 co-migrates in one case but not the other, the authors have now acknowledged this and offer a potential explanation in the discussion. Lastly they adequately addressed and re-analyzed whether EB1 has an effect on initiation probability. The new histograms with the smaller bin sizes now seems to be more in line with what was originally suggested by their initial 1-cdf graphs.

2nd Revision - authors' response

04 September 2017

Response to the reviewer:

The authors thank the reviewer for the positive evaluation.

Additional changes:

We re-analysed the data used to produce fluorescence intensity profiles with an improved script, added some description to the Method and updated the corresponding statistical information in the legends. The new analysis confirmed the profiles as shown in the previous manuscript with the exception of the profiles shown in Fig. EV2C, which required a modification of the corresponding claim in the Results. An additional citation was inserted to support the modified interpretation.

Polarity marks of kymographs in Figure EV5 were removed, because they were incorrect, and the kymographs were flipped for consistency with the other figures in the manuscript.

We added Nicholas I. Cade as an author for his data analysis contribution (fluorescence profiles).

Corresponding Author Name:	Thomas Surrey
Journal Submitted to:	THE EMBO JOURNAL
Manuscript Number:	EMBOJ-2017-97077